# LagEncoder: A Non-Parametric Method for Representation Learning

## Abstract

Non-parametric encoders offer advantages in interpretability and generalizability. However, they often perform significantly worse than deep neural networks on many challenging recognition tasks, and it remains unclear how to effectively apply these techniques to such tasks. In this work, we view all AI recognition tasks as function approximation problems and introduce LagEncoder, a non-parametric, training-free feature extraction method based on finite element basis functions. Our encoder features a universal architecture that can be applied to various types of raw data and recognition tasks. We found that LagEncoder effectively overcomes the limitations of neural networks in regression problems, particularly when fitting multi-frequency functions. The LagEncoder-based model converges quickly and requires low training costs, as only the head is trained. Additionally, LagEncoder provides a parameter-efficient fine-tuning approach. Our experiments on the ImageNet-1K and WikiText dataset demonstrate that pre-trained models using LagEncoder achieve performance improvements within just one training epoch. Furthermore, it does not require adjustments to the original training recipe, extra training data, and the model's total parameters remain nearly unchanged. Our evaluation of the scaling law for model performance indicates that using the LagEncoder is more cost-effective than increasing the model size.

## 1 Introduction

Neural networks have played a pivotal role in the evolution of artificial intelligence, particularly excelling in challenge recognition tasks, where their performance has, in some cases, surpassed human-level capabilities. Furthermore, advances in transfer learning have demonstrated that neural network encoders exhibit notable domain adaptation properties, allowing them to generalize across disparate tasks and data distributions. For instance, an encoder pre-trained for an image classification task can be effectively transferred to an object detection task, even when the underlying data distributions differ, while still maintaining robust feature extraction capabilities.

However, traditional machine learning models often demonstrate better domain adaptation. They provided such feature extractors that require no training and are independent of specific recognition tasks (Pearson, 1901; Sparck Jones, 1972). For example, the kernel functions used in Support Vector Machines (SVM) are unrelated to data labels, resulting in non-trainable feature extractors that inherently provide complete domain adaptation. While traditional models typically offer higher interpretability, they often perform significantly worse than neural networks in specific, highly challenging recognition tasks. It is worth noting that many neural network architectures are based on mathematical methods that do not depend on labeled data. For instance, the linear layer and the attention layer rely on inner products for feature extraction, where the vector basis in linear algebra is fixed. Similarly, convolution layers perform feature extraction through convolution, in the Fourier transform, the kernel function is predefined and independent of the target.

*We view all recognition tasks as function approximation problems and focus on constructing an encoder with complete domain adaptation for challenging tasks.* Through our extensive experiments, we found that, as Fourier transforms, neural networks often struggle to fit multi-frequency functions effectively (see Section 3.1.1). Specifically, when fitting functions with sharp transitions or localized features, a large number of Fourier terms may be required, which can reduce computational efficiency and potentially lead to overfitting. Similarly, neural networks often need to be much

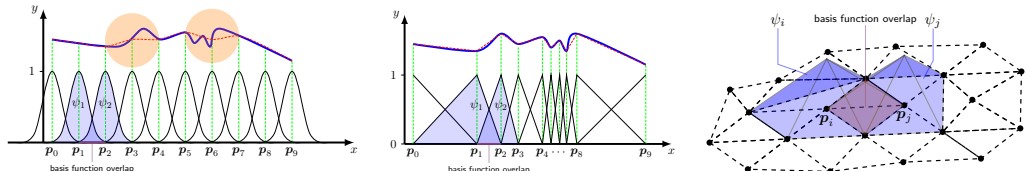

Figure 1: Left - The $F(\boldsymbol{x})$ (solid blue line) is approximated with $f(\boldsymbol{x};\boldsymbol{\theta})$ (dashed red line), which is a linear combination of linear basis functions ($\psi_i$ is represented by the solid black lines). All ten spline basis functions are distributed on a uniform grid, resulting in poor fitting for the orange high-frequency region. Middle - All ten Lagrange basis functions are distributed on a multi-scale grid, resulting in a good fitting over the whole region. Right - Tent-shaped linear basis functions that have a value of 1 at the corresponding node and zero on all other nodes. Two base functions that share two elements have a basis function overlap.

deeper, resulting in a significant increase in model size, to fit such functions properly. In numerical simulation problems, Finite Element Method (FEM) can be used to fit more complex functions and is widely applied in fields such as engineering and mathematical modeling. Because FEM's local basis functions can easily handle multi-scale problems, allowing it to represent features at varying scales, from coarse to fine resolution (see Fig. 1, middle), making it a powerful tool for extracting multi-scale features. In this work, we employed the Lagrange basis function from the FEM as a feature extractor, which we call LagEncoder.

Compared to other encoders, LagEncoder is parameter-free and provides a universal architecture applicable across various recognition domains, such as regression, image classification, image super-resolution, and language modeling. Moreover, LagEncoder serves as an efficient parameter-efficient fine-tuning (PEFT) approach (see Fig. 3 (c)). Our experiments demonstrate that pre-trained models with LagEncoder achieve performance improvements within a single training epoch (*e.g.*, ResNet-50 with an additional 1M parameters achieves a +0.2% gain in validation accuracy in just 40 minutes of fine-tuning on four GPUs). Our evaluation of the scaling law for model performance reveals that LagEncoder is more cost-effective than simply scaling the model. Unlike data augmentation and other PEFT methods, LagEncoder is neither task- nor architecture-specific, requiring no adjustments to the raw training recipe or extra training data to enhance performance.

## 2 METHOD

In general, neural networks provide a mapping from input to predicted output. The classical linear interpolation method includes an error bound formula that demonstrates an explicit relationship between the number of parameters and the mean absolute error. In the context of language models, this relationship is now commonly known as the scaling law (Kaplan et al., 2020). We now consider using this interpolation method to approximate the latent function $F(\boldsymbol{x})$:

$$f(\boldsymbol{x};\boldsymbol{\theta}) = \sum_i \boldsymbol{\theta}_i \cdot \psi_i(\boldsymbol{x}), \quad |f(\boldsymbol{x};\boldsymbol{\theta}) - F(\boldsymbol{x})| \leq \max_{\boldsymbol{\xi} \in \Omega} \|\nabla f(\boldsymbol{\xi})\| \cdot h. \quad (1)$$

where $\boldsymbol{\theta}$ represents the model parameters, $\psi_i(\boldsymbol{x})$ is a basis function, and $f(\boldsymbol{x};\boldsymbol{\theta})$ is the predicted output given input $\boldsymbol{x}$. Here, $h = \max_i \max_j \mathbf{1}_{\text{dist}(\boldsymbol{p}^{(i)}, \boldsymbol{p}^{(j)})=1} \cdot \|\boldsymbol{p}^{(i)} - \boldsymbol{p}^{(j)}\|$ represents the maximum length of mesh edges (see Section 2.1) and $\Omega$ is the domain over which the approximation occurs.

Directly replacing neural networks with traditional interpolation methods is not appropriate. Fig. 1 (left) illustrates the approach of KAN (Liu et al., 2024), which approximates the latent function using a linear combination of spline basis functions. However, when the basis functions are distributed on a uniform grid, the model performs poorly in high-frequency regions. KAN uses Feynman equations as a benchmark; while these equations are complex in expression, they represent very smooth curves. Conversely, when attempting to fit multi-frequency functions, such as $y = \sin(1/x)$, neural networks are likely to outperform KAN due to their flexibility.

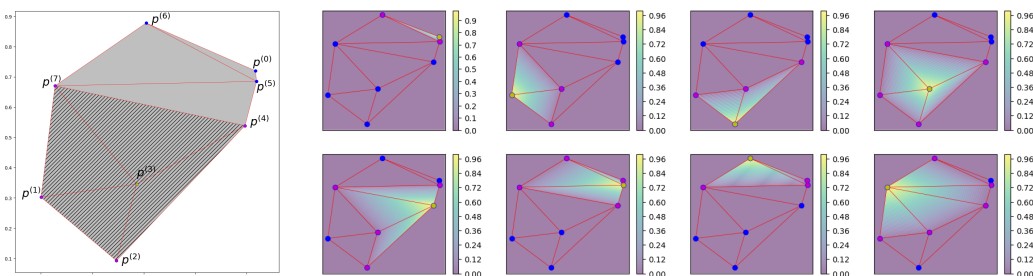

Figure 2: Left - Mesh with eight nodes and seven triangles. Right - Contours of eight Lagrange basis functions, linear variation of $\psi_i$ associated with node $\boldsymbol{p}^{(i)}$ across all triangles.

Mathematically, the FEM offers a standard solution to multi-frequency and high-dimensional challenges. As shown in Fig. 1 (middle), FEM adaptively allocates more parameters to high-frequency regions and fewer to flat regions, enhancing performance in capturing complex multi-frequency behavior. Fig. 1 (middle) illustrates ten basis Lagrange functions for a one-dimensional input space, while the right side shows two basis Lagrange functions for a two-dimensional input space. These tent-shaped linear basis functions take a value of 1 at their corresponding node and 0 at all others. In Section 2.2, we construct the Lagrange basis functions $\psi_i(\boldsymbol{x})$ using FEM. In Section 2.4, we propose two domain decomposition methods to effectively estimate the basis function distribution by analyzing the dataset.

## 2.1 MESH

In the context of FEM, elements often serve as the fundamental building blocks of the triangulation *mesh*, taking the form of *simplices* created by connecting *nodes*. For instance, in 1D FEM, simplices are intervals (see Fig. 1, left); in 2D FEM, triangles with three nodes are commonly used (see Fig. 1, right), while 3D FEM often employs tetrahedra with four nodes. This concept is visually depicted in Fig. 2 (left), where the mesh consists of eight nodes and seven triangles. This type of mesh is established by specifying the coordinates of discrete nodes and the vertex indices of simplices. Let $d$ represent the dimensions, $\{\boldsymbol{p}^{(i)}\}_{i=0}^{n-1}$ denote the grid nodes, and introduce a matrix $\boldsymbol{P}$ to store the node coordinates:

$$\boldsymbol{P}_{i,j} = \boldsymbol{p}_j^{(i)}.$$

Additionally, utilize a matrix $\boldsymbol{T}$ to store the indices of nodes constituting the simplices within the triangulation. Specifically, access the $j$-th sorted vertex of the $i$-th simplex in this mesh as $\boldsymbol{P}_{\boldsymbol{T}_{i,j},:}$. Fig. 2 (left) illustrate a matrix $\boldsymbol{T}$ takes the following form:

$$\boldsymbol{T} = \begin{bmatrix} 6 & 0 & 3 & 2 & 7 & 3 & 4 \\ 7 & 6 & 7 & 3 & 4 & 4 & 3 \\ 5 & 5 & 1 & 1 & 5 & 7 & 2 \end{bmatrix}^T.$$

This matrix serves to describe all seven simplices within the mesh, such as the first simplex $\triangle \boldsymbol{p}^{(6)} \boldsymbol{p}^{(7)} \boldsymbol{p}^{(5)}$ and the last simplex $\triangle \boldsymbol{p}^{(4)} \boldsymbol{p}^{(3)} \boldsymbol{p}^{(2)}$.

The first-order Lagrange basis studied in this article, denoted as $\{\psi_0(\boldsymbol{x}), \cdots, \psi_{n-1}(\boldsymbol{x})\} \subset P_1(\mathbb{R}^d)$, are piecewise linear polynomials associated with nodes $\{\boldsymbol{p}^{(0)}, \cdots, \boldsymbol{p}^{(n-1)}\}$. These functions are defined such that $\psi_i(\boldsymbol{p}^{(j)}) = \mathbf{1}_{i=j}$. Fig. 2 (right) illustrates this: $\psi_i(\boldsymbol{x})$ corresponds to node $\boldsymbol{p}^{(i)}$, exhibiting linear variation across all elements. Its support encompasses the union of all neighboring elements of node $\boldsymbol{p}^{(i)}$ (refer to Appendix B for a 3-dimensional visualization). For example, $\text{supp}(\psi_3) = \triangle \boldsymbol{p}^{(3)} \boldsymbol{p}^{(4)} \boldsymbol{p}^{(7)} \cup \triangle \boldsymbol{p}^{(3)} \boldsymbol{p}^{(7)} \boldsymbol{p}^{(1)} \cup \triangle \boldsymbol{p}^{(3)} \boldsymbol{p}^{(1)} \boldsymbol{p}^{(2)} \cup \triangle \boldsymbol{p}^{(3)} \boldsymbol{p}^{(2)} \boldsymbol{p}^{(4)}$.

## 2.2 LAGENCODER

Now, we formulate the Lagrange basis from its original definition to establish the foundational architecture of LagEncoder. It is important to highlight that the traditional Lagrange basis involves unbalanced computing of barycentric coordinates, which may not be well-suited for parallel deep

learning platforms (see Appendix C for details on the traditional definition of the Lagrange basis). Consequently, in this subsection, we re-derive the Lagrange basis to enhance parallel computing.

Let $n_t$ represent the number of simplices in the multiscale mesh. We introduce the Parameters Tensor $\mathsf{S}$ defined as:

$$
\mathsf{S}_{j,:,:} = \begin{bmatrix} \boldsymbol{p}_0^{(\boldsymbol{T}_{j,0})} & \cdots & \boldsymbol{p}_{d-1}^{(\boldsymbol{T}_{j,0})} & 1 \\ \vdots & \ddots & \vdots & \vdots \\ \boldsymbol{p}_0^{(\boldsymbol{T}_{j,d-1})} & \cdots & \boldsymbol{p}_{d-1}^{(\boldsymbol{T}_{j,d-1})} & 1 \\ \boldsymbol{p}_0^{(\boldsymbol{T}_{j,d})} & \cdots & \boldsymbol{p}_{d-1}^{(\boldsymbol{T}_{j,d})} & 1 \end{bmatrix}^{-1}, \quad j = 0, \cdots, n_t - 1.
$$

Additionally, we introduce the Node Membership tensor $\mathsf{M}$ defined as:

$$
\mathsf{M}_{i,j,k} = \begin{cases} 1, & \text{if the } i\text{-th node matches the } k\text{-th vertex of the } j\text{-th simplex,} \\ 0, & \text{other cases.} \end{cases}
$$

By defining:

$$
\boldsymbol{U}_{j,k}(\boldsymbol{x}) = \sum_{\tau=0}^{d-1} \mathsf{S}_{j,\tau,k} \cdot \boldsymbol{x}_\tau + \mathsf{S}_{j,d,k}, \quad j = 0, \cdots, n_t - 1, \ k = 0, \cdots, d.
$$

We will demonstrate in Appendix A that the following function qualifies the definition of Lagrange basis:

$$
\psi_i(\boldsymbol{x}) = \frac{\sum_{j=0}^{n_t-1} \sum_{k=0}^{d} \mathbf{1}_{\min_\tau \boldsymbol{U}_{j,\tau}(\boldsymbol{x}) \geq 0} \cdot \mathsf{M}_{i,j,k} \cdot \boldsymbol{U}_{j,k}(\boldsymbol{x})}{\max\left(\sum_{j=0}^{n_t-1} \sum_{k=0}^{d} \mathbf{1}_{\min_\tau \boldsymbol{U}_{j,\tau}(\boldsymbol{x}) \geq 0} \cdot \mathsf{M}_{i,j,k}, 1\right)}, \quad i = 0, \cdots, n - 1. \tag{2}
$$

So far, we have successfully constructed :

$$
\begin{aligned} \text{LagEncoder} : &\mathbb{R}^d \to [0,1]^n, \\ &\boldsymbol{x} \mapsto (\psi_0(\boldsymbol{x}), \cdots, \psi_{n-1}(\boldsymbol{x})). \end{aligned}
$$

The above basis is in the format of $P_1(\mathbb{R}^d)$, which is very useful for low-dimensional regression tasks (see Section 3.1). Furthermore, if Eq. (1) is a decoupled system, we can decompose the input space into a direct sum of $d$ one-dimensional subspaces. In this case, Eq. (2) simplifies to the Lagrange basis $P_1(\mathbb{R}^1)$:

$$
\psi_i(\boldsymbol{x}) = \min\left(\frac{\text{ReLU}(\boldsymbol{x} - p_{i-1})}{p_i - p_{i-1}}, \frac{\text{ReLU}(p_{i+1} - \boldsymbol{x})}{p_{i+1} - p_i}\right), \quad i = 0, 1, \cdots, n - 1. \tag{3}
$$

The $P_1(\mathbb{R}^1)$ basis has exceptionally low computational complexity, making it well-suited for recognition tasks on large-scale datasets. In the next section, we will introduce the specific approach in detail. We use the $P_1(\mathbb{R}^1)$ basis to implement an adaptive method for learning a decoupled residual system. This method can enhance the performance of pre-trained models on large datasets.

### 2.3 PEFT-LAGENCODER

LagEncoder adheres to the universal approximation theorem and exhibits strong interpretability for low-dimensional recognition tasks (see Section 3.1). However, as the data dimension $d$ increases, the output dimension of LagEncoder grows factorially ($O(d!)$) as described in Eq. (2). This exponential growth makes it computationally expensive for high-dimensional data, such as large images, posing challenges in resource-constrained environments.

To mitigate this, LagEncoder can be integrated as a module in PEFT methods. Consider a perfect model $F(\boldsymbol{x})$ with 100% test accuracy and a pre-trained model $f(\boldsymbol{x}; \boldsymbol{\theta})$. If the pre-trained model already achieves high test accuracy, the residual $F(\boldsymbol{x}) - f(\boldsymbol{x}; \boldsymbol{\theta})$ will be very flat, typically close to zero for different inputs $\boldsymbol{x}$. This sparsity in the residual's support set implies that dimensionality reduction on the pre-trained model's features is likely reversible, enabling the effective application of LagEncoder.

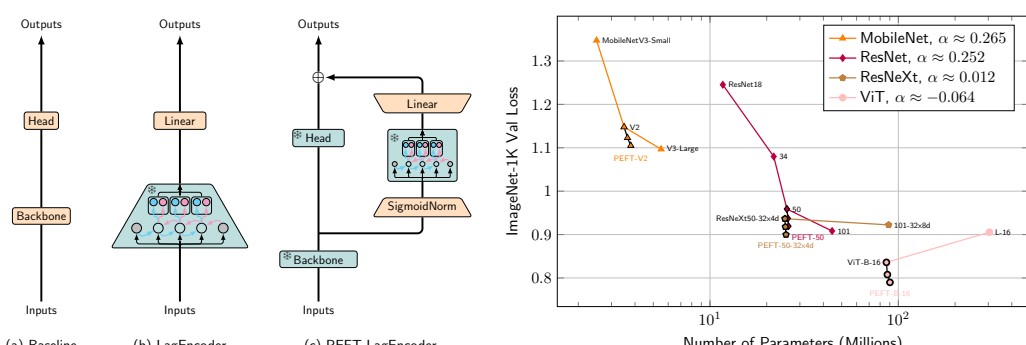

(a) Baseline    (b) LagEncoder    (c) PEFT-LagEncoder

Figure 3: **Left:** (a) A plain network; (b) LagEncoder with a trainable linear layer; (c) LagEncoder applied to PEFT, where the pre-trained model is frozen, and the SigmoidNorm layer for normalization can be trainable or frozen, while the linear layer is trainable. **Right:** Colored lines represent pre-trained models at different scales, and black lines represent our adaptive method, which outperforms model scaling with minimal changes to model size.

In Fig. 3 (c), the SigmoidNorm layer reduces the dimensionality and normalizes the output representation vectors $\boldsymbol{u}$ from the pre-trained backbone:

$$\boldsymbol{v} \leftarrow \mathrm{Sigmoid}(\mathrm{PCA}(\boldsymbol{u})).$$

Here, $\boldsymbol{v}$ represents the reduced-dimensionality feature. The PCA model can be generated using standard unsupervised methods or by training. The $P_1(\mathbb{R}^1)$ LagEncoder is then applied to compute the residual, which is combined with the pre-trained model to estimate the prediction:

$$F(x) \approx f(\boldsymbol{x};\boldsymbol{\theta}) + \sum_i \boldsymbol{\theta}_i^{(\mathrm{linear})} \cdot \psi_i(\boldsymbol{v}). \tag{4}$$

Since the residual $F(x) - f(\boldsymbol{x};\boldsymbol{\theta})$ is sparsely distributed, the additional branch converges quickly during training. Experiments demonstrate that this method minimally increases model size while significantly improving performance within one epoch of training (see Table 2). This approach effectively fine-tunes the model while conserving computational resources.

## 2.4 MULTISCALE DOMAIN DECOMPOSITION METHOD

In our earlier discussion, we introduced the linear interpolation and its associated error-bound formula Eq. (1). However, this tool is not appropriate for machine learning modeling, since we face a crucial challenge: the "given function $F(\boldsymbol{x})$ to be fitted" represents a ground truth that remains unknown. Instead, in the scenario of machine learning, a typical dataset provides us with a collection of input-target pairs. For any given simplex, select a subset $\{(x^{(k_0)}, y^{(k_0)}), \cdots, (x^{(k_{m'-1})}, y^{(k_{m'-1})})\}$ from the training set $\{(x^{(0)}, y^{(0)}), \cdots, (x^{(m-1)}, y^{(m-1)})\}$ where $m'$ is the cardinality of subset, $m$ is the cardinality of subset, $\{k_i\}_{i=0}^{m'-1} \subseteq \{i\}_{i=0}^{m-1}$, and all subset elements reside within the given simplex. Our goal now is to assess the error of $f(\boldsymbol{x};\boldsymbol{\theta})$ within this simplex.

Crucially, due to the linearity of basis functions $\{\psi_0(\boldsymbol{x}), \cdots, \psi_{n-1}(\boldsymbol{x})\}$ within each simplex, their linear combination also remains linear within these simplices. As a result, in the given simplex, there exists a set of coefficients $\boldsymbol{\beta}$ that:

$$\sum_k \boldsymbol{\theta}_k \cdot \psi_k(\boldsymbol{x}^{(i)}) = \boldsymbol{\beta}_0 \boldsymbol{x}_0^{(i)} + \cdots + \boldsymbol{\beta}_{d-1} \boldsymbol{x}_{d-1}^{(i)} + \boldsymbol{\beta}_d.$$

Therefore we can obtain the following error bound by solving a *Ordinary Least Squares* problem

$$\sum_{i=0}^{m'-1} \left| y^{(i)} - \sum_k \boldsymbol{\theta}_k \cdot \psi_k(\boldsymbol{x}^{(i)}) \right|^2 \leq \sum_{i=0}^{m'-1} |y^{(i)} - (\hat{\boldsymbol{\beta}}_0 \boldsymbol{x}_0^{(i)} + \cdots + \hat{\boldsymbol{\beta}}_{d-1} \boldsymbol{x}_{d-1}^{(i)} + \hat{\boldsymbol{\beta}}_d)|^2, \tag{5}$$

where $\hat{\boldsymbol{\beta}} = (\boldsymbol{X}^T \boldsymbol{X})^{-1} \boldsymbol{X}^T \boldsymbol{y}$, $\boldsymbol{X}_{i,:d} = \boldsymbol{x}^{(i)}$, $\boldsymbol{X}_{i,d} = 1$, and $\boldsymbol{y}_i = y^{(i)}$. This formula shows the error bound reduced to 0 when $m' \leq d + 1$. By combining this conclusion with the global error-bound formula Eq. (1), we can summarize two critical goals for mesh generation in modeling:

---

**Algorithm 1** Domain decomposition method of generating multiscale mesh.

---

**Input:** Maximum degrees of freedom $N$ to perform. Depending on the size of the training set.
**Input:** Initial Simplex Indices Matrix $T$ of shape $(1, d + 1)$ with $T_{0,i} = i$.
**Input:** Initial Node Matrix $P$ containing coordinates of $d + 1$ points forming a simplex covering all training raw data.
**Output:** The updated Node Matrix $P$ and Simplex Indices Matrix $T$ of the refined mesh.
- $n \leftarrow d + 1$
**while** $n < N$ **do**
- Create the binary Longest Edge Matrix $M$ where $M_{i,j} = 1$ indicates that the $i$-th edge is the longest side of the $j$-th simplex.
- Formulate the binary Edge Membership Matrix $E$ where $E_{i,j} = 1$ indicates that the $i$-th edge is a side of the $j$-th simplex.
- Establish the binary Data-Simplex Membership Matrix $B$ where $B_{i,j} = 1$ signifies that the $i$-th raw data falls within the $j$-th simplex.
- Compute the index of the priority edge:

$$\arg\min_i \frac{\sum_j \sum_k M_{i,j} B_{k,j}}{\max(\sum_j \sum_k E_{i,j} B_{k,j}, 1)}.$$

  The priority edge is the longest side among many simplices, and these relevant simplices cover a substantial portion of the raw data.
- Insert a new node at the midpoint of the priority edge and update the Node Matrix $P$.
- Update the Simplex Indices Matrix $T$ and utilize it to update mesh edges.
- $n \leftarrow n + 1$
**end while**

---

**Algorithm 2** V-Cycle.

---

**Require:** The pre-trained model $f(x; \theta)$ with frozen parameter $\theta$.
**Require:** Initial parameter $\theta^{(\text{linear})}$ of the linear head.
**Require:** Top-K threshold, $K_1$ and $K_2$ (Suggested defaults: $0.1 \times \text{batch}_{\text{size}}$ and 1 respectively).
**Require:** Mesh updating frequency $N$.
- $k \leftarrow 1$
- **Generate PDF**: Construct a histogram on the interval $[-1, 1]$ to represent the Probability Density Function (PDF) of the empirical data distribution, initially using equal-width binning. The grid $P$ contains $n - 1$ bins, with $n$ coarse nodes.
**while** stopping criterion not met **do**
- Sample a minibatch of $m$ examples from the training set $\{x^{(1)}, \cdots, x^{(m)}\}$ with targets $y^{(i)}$.
- Compute corresponding losses: $l_i \leftarrow L(f(x; \theta) + \sum_i \theta_i^{(\text{linear})} \cdot \psi_i(v^{(i)}), y^{(i)})$.
- **Generate PDF**: Estimate a temporary PDF using the current $m$ examples by selecting the top $K_1$ examples with the highest losses and calculating their proportion in each bin.
- **Update PDF**: Update the PDF using the temporary PDF and Exponential Moving Average.
**if** $k \mod N \equiv 0$ **then**
- **Mesh Refinement**: Mark the top $K_2$ bins as coarse elements and add their midpoints as fine nodes, increasing degrees of freedom to $n + K_2$.
- **Interpolation**: Use linear interpolation to update $\theta^{(\text{linear})}$ at each fine node.
- **Mesh Coarsening**: Solve a system of equations to transform the current histogram back into equal-width binning, reducing the grid's degrees of freedom back to $n$.
- **Interpolation**: Use linear interpolation to update $\theta^{(\text{linear})}$ at each new node.
**end if**
$k \leftarrow k + 1$
**end while**

---

1. Each simplex in the mesh should ideally contain as few original data points from the training set as possible. When each simplex covers no more than $d + 1$ raw data examples, the model perfectly fits the training set.

2. Decreasing the bound of mesh edge lengths results in a reduced bound of error.

Algorithm 1 generates a multiscale mesh, serving as the initial step in constructing the LagEncoder architecture. In each iteration, it adds a new fine node to the grid and subdivides coarse simplices into finer ones. Fig. 7 (left) in Appendix D illustrates this mesh refinement process. For PEFT methods, Algorithm 1 can be replaced by the simpler V-cycle Algorithm 2 from FEM, which updates the mesh by estimating the empirical distribution of training examples. The V-cycle allocates more nodes in high-frequency (dense) regions and fewer in low-frequency (sparse) regions. A visualized workflow is shown in Appendix D (Fig. 7, right).

## 3 EXPERIMENTS

In this section, we present a comprehensive series of experiments to showcase the effectiveness and universality of LagEncoder across various tasks. Our exploration begins with an analysis of its performance in regression tasks, followed by examinations in image and text recognition. Finally, we apply it to the PEFT method and assess its performance on large datasets.

### 3.1 REGRESSION TASKS

#### 3.1.1 THE LIMITATIONS OF TRADITIONAL REGRESSORS AND NEURAL NETWORKS

Traditional regressors often struggle with overfitting, while neural networks address this but perform poorly when fitting multi-frequency functions, a phenomenon known as the *frequency principle* (Xu et al., 2019). As shown in Fig. 4, Support Vector Regression (Platt et al., 1999) fits dataset $\mathbb{B}$ but overfits on dataset $\mathbb{A}$, whereas neural networks perform well on $\mathbb{A}$ but underfit on $\mathbb{B}$. *Neither MLPs, CNNs, nor Transformers can effectively fit sharp transitions in dataset $\mathbb{B}$, often treating these regions as noise to avoid overfitting.* Importantly, no existing model can fit the challenging dataset $\mathbb{C}$.

LagEncoder addresses this gap by combining the strengths of traditional regressors and neural networks. It leverages a multiscale mesh to fit high-frequency regions using fine simplices adaptively. Fig. 4 (right) demonstrates the LagEncoder-based model's unique ability to handle dataset $\mathbb{C}$, a task no other model can achieve. Appendix F.3 (Fig. 9) further illustrates its gradual fitting process during training. Comparison experiments in Appendix F.1 show that the LagEncoder-based model consistently achieves high $R^2$ scores across all test sets, demonstrating its robustness and effectiveness, even in high-noise and multi-frequency scenarios.

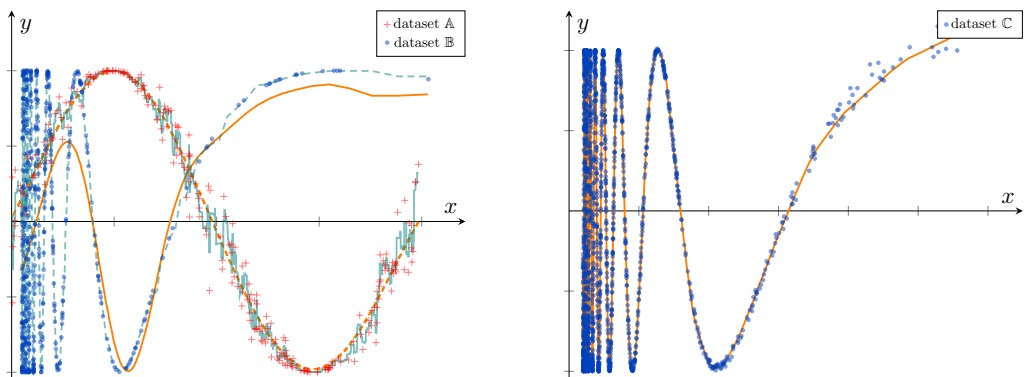

Figure 4: Left - Performance of traditional regressors and neural networks on high-noise dataset $\mathbb{A} = (x,y)|X \sim u(0, \frac{\pi}{4}), Y \sim N(\sin 8x, |\cos 8x|)$ and multi-frequency dataset $\mathbb{B} = (x,y)|\frac{1}{X} \sim u(0.02, 0.5), y = \sin \frac{1}{x}$. The dashed teal curve shows that traditional regressors (e.g., Support Vector Regression) succeed on $\mathbb{B}$ but overfit $\mathbb{A}$ (solid teal curve). Conversely, neural networks (e.g., Multi-Layer Perceptron) fit $\mathbb{A}$ well (dashed orange curve) but underfit $\mathbb{B}$ (solid orange curve). Right - The LagEncoder-based model demonstrates exceptional adaptability, successfully fitting dataset $\mathbb{C} = (x,y)|\frac{1}{X} \sim u(0.02, 0.5), Y \sim N(\sin \frac{1}{x}, 0.5x^2)$, which combines high noise and multi-frequency features.

### 3.1.2 SCALING LAW AND ERROR-BOUND FORMULA

Our LagEncoder demonstrates strong interpretability, supported by the universal approximation theorem and our quantitative experimental results. In a triangular mesh, the number of parameters $N$ in the model's linear head is proportional to the number of simplices $n_t$ in the mesh. Since $(n_t/d!)^{1/d} = O(h^{-1})$, the error bound formula in Eq. (1) is derived as:

$$L(N) = |f(\boldsymbol{x}; \boldsymbol{\theta}) - F(\boldsymbol{x})| = O(\max_{\boldsymbol{\xi} \in \Omega} \|\nabla f(\boldsymbol{\xi})\| \cdot n_t^{-1/d}) = O(N^{-1/d}). \tag{6}$$

The experiment shown in Fig. 5 demonstrates that we perfectly predicted the empirical scaling law.

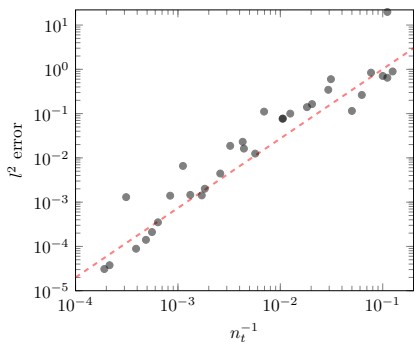 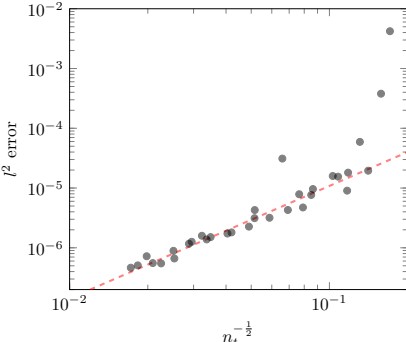

Figure 5: Left - Results of 32 experiments fitting the 1-dimensional function $y = \sin \frac{1}{x}$, where each gray point represents an experiment, showing the relationship between $n_t^{-1}$ and Mean Square Error (MSE). Right - Results of 32 experiments fitting the 2-dimensional function $y = \sum_{i=1}^{2} \sin \frac{\boldsymbol{x}_i}{2\pi}$, with gray points illustrating individual experiments and the relationship between $n_t^{-1/2}$ and MSE.

### 3.2 NATURAL LANGUAGE PROCESSING

In this section, we explored the practical application of LagEncoder for text feature extraction on the AG News dataset (Zhang et al., 2015) for classification tasks. The degree of freedom $n$ of LagEncoder was set to 64. Using the SGD optimizer, we minimized the cross-entropy loss with an initial learning rate of 5.0, reduced by a factor of 0.1 every two epochs, and a batch size of 32. The experiment achieved 90.01% test accuracy after the first epoch and 90.4% within five epochs. A unique feature of the LagEncoder-based model in text classification is that its parameter count is independent of the token count. Compared to word2vec-based networks, which require over 6.13 million parameters, the LagEncoder-based model achieves comparable classification performance with only 256 parameters, offering a significant reduction in complexity.

Through experiments, we found three key limitations of existing PEFT methods:

- Dependence on Transfer Learning: Methods like LoRA struggle to outperform pre-trained models on the same or similar datasets, as no domain adaptation is needed. Additional training data is often required for these methods to be effective, as shown in Table 1. Their strength lies mainly in transfer learning scenarios. However, when trained from scratch on the raw dataset, they fail to outperform pre-trained models (see Tables 2 and 5).
- Sensitive Training Requirements: PEFT methods require specific training recipes, such as small learning rates. Without these, performance often deteriorates from the first epoch.
- Task and Architecture Limits: Many PEFT methods, like LoHA (Hyeon-Woo et al., 2021) and IA3 (Liu et al., 2022), are restricted to specific tasks or architectures, such as Conv1D or linear layers.

We compare fine-tuning (FT), LoRA (Hu et al., 2021), and LagEncoder on WikiText benchmarks (Merity et al., 2016) using GPT2 (Radford et al., 2019) for causal language modeling and RoBERTa Liu (2019) for masked language modeling. Pre-trained models and the default random seed (42)

of the HuggingFace Transformers library (Wolf et al., 2020) are used. A reduced learning rate ($\leq 10^{-5}$) is applied for LoRA to prevent performance degradation in these experiments. As shown in Table 1, our method overcomes the mentioned challenges, performing well even without extra training data. It is also versatile for transfer learning (see Table 5).

| Dataset | Method | GPT2 | | | GPT2-Medium | | | GPT2-Large | | | Roberta-Base | | |
|---|---|---|---|---|---|---|---|---|---|---|---|---|---|
| | | #Params | PPL | Seq/s | #Params | PPL | Seq/s | #Params | PPL | Seq/s | #Params | PPL | Seq/s |
| Wikitext-2 | FT | 124.4M | 21.6623 | 7.36 | 354.8M | 16.1183 | 3.38 | 774.0M | 13.9927 | 1.71 | 124.7M | 3.6337 | 9.82 |
| | LoRA | 0.295M | 21.6582 | **13.05** | 0.393M | 16.1183 | **6.00** | 0.737M | 13.9973 | 1.69 | 0.295M | 3.6347 | 9.68 |
| | LagEncoder | 0.203M | **21.597** | 9.76 | 0.404M | **16.1086** | 5.83 | 0.607M | **13.9964** | **3.02** | 0.303M | **3.6346** | **15.39** |
| Wikitext-103 | FT | 124.4M | 21.4234 | 6.98 | 354.8M | 15.8900 | 3.50 | 774.0M | 13.8468 | 1.58 | 124.7M | 3.6415 | 10.05 |
| | LoRA | 0.295M | 21.4188 | **12.90** | 0.393M | 15.8873 | 6.29 | 0.737M | 13.8486 | 1.67 | 0.295M | 3.6425 | 9.85 |
| | LagEncoder | 0.203M | **21.3401** | 9.58 | 0.404M | **15.8674** | **6.66** | 0.607M | **13.8443** | **2.52** | 0.303M | **3.6414** | 12.37 |

Table 1: GPT2 and RoBERTa with Different Adaptation Methods on the WikiText Benchmark. We report the number of trainable parameters, perplexity (PPL, lower is better), and training throughput (sequences per second) for language modeling tasks. For a fair comparison, we adjusted the number of parameters of our method to be similar to those of LoRA.

## 3.3 COMPUTER VISION

As described in Section 2.3, LagEncoder can serve as a non-parametric module for the PEFT method. The new model includes a residual branch where the SigmoidNorm layer and linear head contain trainable parameters (see Fig. 3 (c)). In this section, we illustrate the effectiveness of PEFT-LagEncoder through ablation studies. To ensure a fair comparison with baseline models, we adopt a stricter experimental setup: *the original training recipe of the pre-trained model cannot be modified when training the residual branch*.

This restriction is crucial to ablation studies because the training recipes for pre-trained models often have room for optimization (Wightman et al., 2021). Changes to batch size, optimizer, weight decay rate, or the order of applying data augmentations could potentially improve the performance of the pre-trained model (Touvron et al., 2019; 2021). To avoid such effects, we selected pre-trained models from TorchVision with publicly available training recipes as our baselines. The weights of these pre-trained models closely reproduce the results from the original papers on the ImageNet-1K dataset (Russakovsky et al., 2015), with recipes available at (TorchVision Contributors, 2024).

| Model | Method | # Trainable Parameters | Acc@1 | Acc@5 | Speed (img/s) | Training Time (total) |
|---|---|---|---|---|---|---|
| MobileNet-V2 | Baseline* | 3.5 M | 71.878 | **90.286** | | 16h 30m |
| | LoRA | 341,360 | 71.686 | 90.280 | 701.51 | 32m 54s |
| | PromptTuning | 324,080 | 71.798 | 90.278 | **864.09** | 32m 12s |
| | LagEncoder | 328,720 | **71.934** | 90.268 | 685.17 | **32m 12s** |
| ResNet-50 | Baseline* | 25.6 M | 76.130 | 92.862 | | 2d 1h 15m |
| | LoRA | 597,933 | 76.288 | 92.972 | **440.18** | **32m 48s** |
| | PromptTuning | 536,750 | **76.674** | **93.128** | 434.07 | 39m 6s |
| | LagEncoder | 540,688 | 76.274 | 92.932 | 411.50 | 40m 42s |
| ResNeXt-50_32x4d | Baseline* | 25.0 M | 77.618 | **93.698** | | 3d 1h 30m |
| | LoRA | 246,830 | 77.450 | 93.580 | **334.72** | **48m 54s** |
| | PromptTuning | 149,552 | 77.424 | 93.602 | 333.71 | 50m 9s |
| | LagEncoder | 135,172 | **77.650** | 93.672 | 315.33 | 51m 9s |
| ViT-B16 | Baseline* | 86.6 M | 81.072 | **95.318** | | 3d 3h 20m |
| | LoRA | 617,942 | 80.914 | 95.282 | 331.02 | **50m 6s** |
| | PromptTuning | 71,760 | 80.770 | 95.172 | **334.21** | 55m 9s |
| | LagEncoder | 67,076 | **81.082** | 95.316 | 316.31 | 56m 54s |

Table 2: Comparison of Adaptation Methods on Various Networks and the ImageNet-1K Dataset. PEFT-LagEncoder demonstrates superior performance compared to other methods, offering competitive trainable parameters, validation accuracy, training throughput (images per second), and total training time (five epochs). * indicates numbers published in prior works.

Due to space constraints, we present only the comparison results in Table 2. In Appendix E, we highlight the strong stability of PEFT-LagEncoder, showing ±0.07% fluctuation in validation accuracy across independent model training (see Table 3). Trainable parameters in our method originate from the SigmoidNorm and linear layer (Fig. 3(c)) and the total number of trainable parameters can be calculated as $N = (C_{in} + 1) \times d + d \times n \times C$, where $N$ is the number of model parameters, $C_{in}$ is the SigmoidNorm input dimension, $d$ the PCA output dimension, $n$ is the mesh node count (degrees of freedom), and $C$ the total output dimension.

Increasing $n$ and $d$ improves model performance (see Appendix E, Table 4). We also study empirical scaling laws for model performance on cross-entropy loss to demonstrate how our adaptation method fully exploits the potential of pre-trained models. (Kaplan et al., 2020) proposed an empirical formula where the loss scales as a power-law with model size:

$$L(N) = \left(\frac{c}{N}\right)^{\alpha}$$

where $L(N)$ is the cross-entropy loss on the validation dataset, $c$ and $\alpha$ are constants related to the model type. Fig. 3 (right) shows that our adaptation method is more effective than simply scaling the model size. Our method increases the $\alpha$ for MobileNet (Howard et al., 2017) from 0.27 to 0.47, for ResNet He et al. (2016) from 0.25 to 2.1, for ResNeXt Xie et al. (2017) from 0.01 to 3.1, and for ViT (Dosovitskiy et al., 2020) from -0.06 to 1.2.

Our method offers significant practical value. Unlike other model scaling and adaptation approaches, it requires no adjustments to the training recipe, has low computational demands with minimal trainable parameters, and converges within a single epoch. For instance, on 4x A6000 GPUs, it adds 1 million parameters to a ResNet-50 model and achieves a 0.2% accuracy gain in just 40 minutes of training. Moreover, unlike other PEFT methods, our approach is not limited to transfer learning and can improve performance directly on the raw dataset (see Table 2).

## 4 FUTURE DIRECTIONS

It is important to recognize that LagEncoder has certain limitations. As described in Section 2.3, the high computational cost of extracting features from high-dimensional data restricts its direct application to large-scale datasets. Currently, we address this by incorporating LagEncoder as part of an adaptation method, but we aim to improve the encoder in the future to be able to extract features directly from large images. Additionally, the experiments in Section 3.3 show that while our adaptation method significantly reduces cross-entropy on the ImageNet-1K validation set and consistently improves validation accuracy, the magnitude of improvement of accuracy is relatively modest, which we intend to explore further. We did not compare LagEncoder with models like SVM or KAN, which offer strong interpretability, because their underlying principles are entirely different. Moreover, we prioritize performance on challenging tasks where most interpretable models often lack relevant experimental results.

## 5 CONCLUSION

We explored domain adaptation in transfer learning and proposed a non-parametric encoder that does not require training. This encoder has a universal design suitable for diverse raw data and tasks. It extracts features by estimating data distribution across a multi-scale grid, enabling strong generalization. Our experiments demonstrated several key advantages of the LagEncoder-based model: 1) **Strong mathematical explainability**: The empirical model scaling law aligns perfectly with the error-bounds formula of linear interpolation, providing a solid understanding of how it learns representations. 2) **Fast training**: With only one linear layer to train, the model typically converges in just one or two epochs, making it ideal for large datasets. 3) **Not limited to transfer learning**: Unlike most PEFT methods, which are task-specific and underperform on raw datasets, our method works effectively without these constraints.

In summary, LagEncoder is a novel encoder rooted in the universal approximation theorem, requiring no extensive training, fine-tuning, or heavy computational resources. It offers an explainable and efficient approach to representation learning, providing an alternative to black-box methods.

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

## A    PROOF OF LAGRANGE BASIS EXPRESSION

We will now demonstrate, in three concise steps, that Eq. (2) qualifies as a Lagrange basis function.
**Piecewise Linear:** Since $\{U_{j,0}, \cdots, U_{j,d}\}$ are piecewise linear functions, their linear combination $L_i$ is also piecewise linear.
**Kronecker Delta:** From the definition of $U$ and $\mathsf{S}$, we have the following equation:

$$\begin{bmatrix} x_0 & \cdots & x_{d-1} & 1 \end{bmatrix} = \begin{bmatrix} U_{j,0}(x) & \cdots & U_{j,d}(x) \end{bmatrix} \begin{bmatrix} p_0^{(T_{j,0})} & \cdots & p_{d-1}^{(T_{j,0})} & 1 \\ \vdots & \ddots & \vdots & \vdots \\ p_0^{(T_{j,d-1})} & \cdots & p_{d-1}^{(T_{j,d-1})} & 1 \\ p_0^{(T_{j,d})} & \cdots & p_{d-1}^{(T_{j,d})} & 1 \end{bmatrix}.$$

Decomposing this equation, we obtain $x = \sum_{k=0}^{d} U_{j,k}(x) p^{(T_{j,k})}$ and $\sum_{k=0}^{d} U_{j,k}(x) = 1$. This implies two important conclusions: $U_{j,k'}(p^{(T_{j,k''})}) = \mathbf{1}_{k'=k''}$ and $\min_\tau U_{j,\tau}(x) \geq 0$ is true if and only if $x$ belongs to the $j$-th simplex. Therefore, we have

$$\begin{cases} \mathbf{1}_{\min_\tau U_{j,\tau}(p^{(i)}) \geq 0} = \mathsf{M}_{i,j,k} = U_{j,k}(p^{(i)}) = 1, & \text{if } i = T_{j,k} \\ \mathsf{M}_{i,j,k} = U_{j,k}(p^{(i)}) = 0, & \text{if } i \neq T_{j,k} \end{cases}$$

This proves that $\psi_i(p^{(j)}) = \mathbf{1}_{i=j}$.
**Globally Continuity:** Lastly, since $\psi_i$ is inherently linear within all simplices and exhibits continuity across all grid nodes, we can conclude that $\psi_i$ is globally continuous.

## B    VISUALIZATION OF LAGRANGIAN BASIS

In section 2, we introduced first-order Lagrange basis functions, a set of piecewise linear functions defined on a mesh. Each basis function corresponds to a node.

Consider the grid depicted in Fig. 6 (left). Taking the node $p^{(20)}$ as an example, it has a total of four neighboring nodes: $p^{(5)}$, $p^{(0)}$, $p^{(7)}$, and $p^{(4)}$. By connecting these nodes, we can determine the support of the basis function $\psi_{20}$.

In Fig. 6 (middle), we present the function graphs of $\psi_{20}$ and $\psi_7$. It can be observed that these functions exhibit linear variations on each mesh triangle. Taking $\psi_{20}$ as an example, its function value at $p_{20}$ is 1, and 0 at all other nodes. Similarly, $\psi_7$ has a function value of 1 at $p_7$ and 0 at other nodes.

In Fig. 6 (right), the orange triangles represent the function graph of $f(x; \theta)$ on the domain $\triangle p^{(0)} p^{(7)} p^{(9)}$, where the function values of $f(x; \theta)$ at the vertices $(p^{(0)}, p^{(7)}, p^{(9)})$ are $(\theta_0, \theta_7, \theta_9)$, respectively. The green dots represent a subset of training set, where the projections (raw data) fall on $\triangle p^{(0)} p^{(7)} p^{(9)}$. As shown in equation (2), when the number of green points does not exceed three, there exists a solution of $(\theta_0, \theta_7, \theta_9)$ such that all green points lie on the surface of $f(x; \theta)$, such that MSE reach a minimum value of 0.

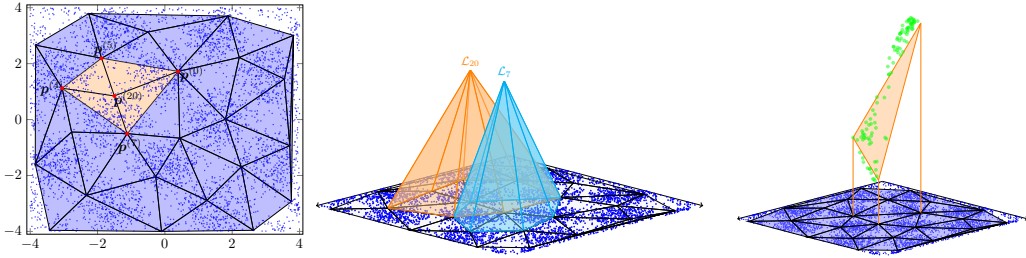

Figure 6: Data visualization. Left - an example of 2-dimensional mesh. Middle - the graphs of basis functions $\psi_{20}$ and $\psi_7$. Right - the graphs of the function $f(x; \theta)$ and a subset of training set.

## C  THE TRADITIONAL EXPRESSION OF LAGRANGE BASIS

Given the complexity of FEM as a numerical method, to enhance understanding, we start with a two-dimensional case and a triangulated mesh to illustrate the traditional expression of basis functions. Let triangle $\triangle$ be defined by nodes $\{\boldsymbol{p}^{(i)}, \boldsymbol{p}^{(j)}, \boldsymbol{p}^{(k)}\}$. The following barycentric coordinates $\{\lambda_{\triangle,i}, \lambda_{\triangle,j}, \lambda_{\triangle,k}\}$ are three first-degree polynomials of $\boldsymbol{x}$

$$\begin{bmatrix} \lambda_{\triangle,i}(\boldsymbol{x}) \\ \lambda_{\triangle,j}(\boldsymbol{x}) \\ \lambda_{\triangle,k}(\boldsymbol{x}) \end{bmatrix} = \begin{bmatrix} \boldsymbol{p}_0^{(i)} & \boldsymbol{p}_0^{(j)} & \boldsymbol{p}_0^{(k)} \\ \boldsymbol{p}_1^{(i)} & \boldsymbol{p}_1^{(j)} & \boldsymbol{p}_1^{(k)} \\ 1 & 1 & 1 \end{bmatrix}^{-1} \begin{bmatrix} \boldsymbol{x}_0 \\ \boldsymbol{x}_1 \\ 1 \end{bmatrix}.$$

Referring to the instance depicted in Fig. 2 (left), the mesh consists of eight nodes and seven triangles. Specifically, let $\{\triangle^{(j)} = \triangle \boldsymbol{p}^{(T_{j,0})} \boldsymbol{p}^{(T_{j,1})} \boldsymbol{p}^{(T_{j,2})} | j = 0, \cdots, 6\}$. We will now verify that the following $\psi_3$ corresponds to the third basis function in this mesh

$$\psi_3(\boldsymbol{x}) = \frac{1}{\max(\sum_{i \in \{2,3,6,5\}} \mathbf{1}_{\boldsymbol{x} \in \triangle^{(i)}}, 1)} \sum_{i \in \{2,3,6,5\}} \mathbf{1}_{\boldsymbol{x} \in \triangle^{(i)}} \lambda_{\triangle^{(i)},3}(\boldsymbol{x}).$$

First, $\psi_3$ possesses values of *Kronecker Delta*:

$$\psi_3(\boldsymbol{x}) = \begin{cases} \frac{1}{\max(0,1)}(0 \cdot \lambda_{\triangle^{(2)},3}(\boldsymbol{x}) + 0 \cdot \lambda_{\triangle^{(3)},3}(\boldsymbol{x}) + 0 \cdot \lambda_{\triangle^{(6)},3}(\boldsymbol{x}) + 0 \cdot \lambda_{\triangle^{(5)},3}(\boldsymbol{x})) = 0, & \text{if } \boldsymbol{x} = \boldsymbol{p}^{(0)}, \\ \frac{1}{\max(2,1)}(1 \cdot 0 \qquad +1 \cdot 0 \qquad +0 \cdot \lambda_{\triangle^{(6)},3}(\boldsymbol{x}) + 0 \cdot \lambda_{\triangle^{(5)},3}(\boldsymbol{x})) = 0, & \text{if } \boldsymbol{x} = \boldsymbol{p}^{(1)}, \\ \frac{1}{\max(2,1)}(0 \cdot \lambda_{\triangle^{(2)},3}(\boldsymbol{x}) + 1 \cdot 0 \qquad +1 \cdot 0 \qquad +0 \cdot \lambda_{\triangle^{(5)},3}(\boldsymbol{x})) = 0, & \text{if } \boldsymbol{x} = \boldsymbol{p}^{(2)}, \\ \frac{1}{\max(4,0)}(1 \cdot 1 \qquad +1 \cdot 1 \qquad +1 \cdot 1 \qquad +1 \cdot 1) \qquad = 1, & \text{if } \boldsymbol{x} = \boldsymbol{p}^{(3)}, \\ \frac{1}{\max(2,1)}(0 \cdot \lambda_{\triangle^{(2)},3}(\boldsymbol{x}) + 0 \cdot \lambda_{\triangle^{(3)},3}(\boldsymbol{x}) + 1 \cdot 0 \qquad +1 \cdot 0) \qquad = 0, & \text{if } \boldsymbol{x} = \boldsymbol{p}^{(4)}, \\ \frac{1}{\max(0,1)}(0 \cdot \lambda_{\triangle^{(2)},3}(\boldsymbol{x}) + 0 \cdot \lambda_{\triangle^{(3)},3}(\boldsymbol{x}) + 0 \cdot \lambda_{\triangle^{(6)},3}(\boldsymbol{x}) + 0 \cdot \lambda_{\triangle^{(5)},3}(\boldsymbol{x})) = 0, & \text{if } \boldsymbol{x} = \boldsymbol{p}^{(5)}, \\ \frac{1}{\max(0,1)}(0 \cdot \lambda_{\triangle^{(2)},3}(\boldsymbol{x}) + 0 \cdot \lambda_{\triangle^{(3)},3}(\boldsymbol{x}) + 0 \cdot \lambda_{\triangle^{(6)},3}(\boldsymbol{x}) + 0 \cdot \lambda_{\triangle^{(5)},3}(\boldsymbol{x})) = 0, & \text{if } \boldsymbol{x} = \boldsymbol{p}^{(6)}, \\ \frac{1}{\max(2,1)}(1 \cdot 0 \qquad +0 \cdot \lambda_{\triangle^{(3)},3}(\boldsymbol{x}) + 0 \cdot \lambda_{\triangle^{(6)},3}(\boldsymbol{x}) + 1 \cdot 0) \qquad = 0, & \text{if } \boldsymbol{x} = \boldsymbol{p}^{(7)}, \end{cases}$$

Then, $\psi_3$ is a first-degree polynomial in every triangle and supp $\psi_3 = \text{inn} \cup_{i \in \{2,3,6,5\}} \triangle^{(i)}$:

$$\psi_3(\boldsymbol{x}) = \begin{cases} \frac{1}{\max(0,1)}(0 \cdot \lambda_{\triangle^{(2)},3}(\boldsymbol{x}) + 0 \cdot \lambda_{\triangle^{(3)},3}(\boldsymbol{x}) + 0 \cdot \lambda_{\triangle^{(6)},3}(\boldsymbol{x}) + 0 \cdot \lambda_{\triangle^{(5)},3}(\boldsymbol{x})) = 0, & \text{if } \boldsymbol{x} \in \text{inn } \boldsymbol{T}_0, \\ \frac{1}{\max(0,1)}(0 \cdot \lambda_{\triangle^{(2)},3}(\boldsymbol{x}) + 0 \cdot \lambda_{\triangle^{(3)},3}(\boldsymbol{x}) + 0 \cdot \lambda_{\triangle^{(6)},3}(\boldsymbol{x}) + 0 \cdot \lambda_{\triangle^{(5)},3}(\boldsymbol{x})) = 0, & \text{if } \boldsymbol{x} \in \text{inn } \boldsymbol{T}_1, \\ \frac{1}{\max(1,1)}(1 \cdot \lambda_{\triangle^{(2)},3}(\boldsymbol{x}) + 0 \cdot \lambda_{\triangle^{(3)},3}(\boldsymbol{x}) + 0 \cdot \lambda_{\triangle^{(6)},3}(\boldsymbol{x}) + 0 \cdot \lambda_{\triangle^{(5)},3}(\boldsymbol{x})) = \lambda_{\triangle^{(2)},3}(\boldsymbol{x}), & \text{if } \boldsymbol{x} \in \text{inn } \boldsymbol{T}_2, \\ \frac{1}{\max(1,1)}(0 \cdot \lambda_{\triangle^{(2)},3}(\boldsymbol{x}) + 1 \cdot \lambda_{\triangle^{(3)},3}(\boldsymbol{x}) + 0 \cdot \lambda_{\triangle^{(6)},3}(\boldsymbol{x}) + 0 \cdot \lambda_{\triangle^{(5)},3}(\boldsymbol{x})) = \lambda_{\triangle^{(3)},3}(\boldsymbol{x}), & \text{if } \boldsymbol{x} \in \text{inn } \boldsymbol{T}_3, \\ \frac{1}{\max(0,1)}(0 \cdot \lambda_{\triangle^{(2)},3}(\boldsymbol{x}) + 0 \cdot \lambda_{\triangle^{(3)},3}(\boldsymbol{x}) + 0 \cdot \lambda_{\triangle^{(6)},3}(\boldsymbol{x}) + 0 \cdot \lambda_{\triangle^{(5)},3}(\boldsymbol{x})) = 0, & \text{if } \boldsymbol{x} \in \text{inn } \boldsymbol{T}_4, \\ \frac{1}{\max(1,1)}(0 \cdot \lambda_{\triangle^{(2)},3}(\boldsymbol{x}) + 0 \cdot \lambda_{\triangle^{(3)},3}(\boldsymbol{x}) + 0 \cdot \lambda_{\triangle^{(6)},3}(\boldsymbol{x}) + 1 \cdot \lambda_{\triangle^{(5)},3}(\boldsymbol{x})) = \lambda_{\triangle^{(5)},3}(\boldsymbol{x}), & \text{if } \boldsymbol{x} \in \text{inn } \boldsymbol{T}_5, \\ \frac{1}{\max(1,1)}(0 \cdot \lambda_{\triangle^{(2)},3}(\boldsymbol{x}) + 0 \cdot \lambda_{\triangle^{(3)},3}(\boldsymbol{x}) + 1 \cdot \lambda_{\triangle^{(6)},3}(\boldsymbol{x}) + 0 \cdot \lambda_{\triangle^{(5)},3}(\boldsymbol{x})) = \lambda_{\triangle^{(6)},3}(\boldsymbol{x}), & \text{if } \boldsymbol{x} \in \text{inn } \boldsymbol{T}_6, \end{cases}$$

Finally, since $\psi_3$ is continuous in all nodes and first-degree in all triangles, it is globally continuous.

## D  VISUALIZATION OF MESH REFINEMENT

In this work, we utilize the triangle mesh for constructing $P_1(\mathbb{R}^d)$ Lagrange basis. As the mesh is refined, each simplex will contain the same number of points, so Algorithm 1 is an equal-frequency binning method in $d$-dimensional space.

(Fig. 7, left) depicts a mesh variety during the iterations of Algorithm 1. In the coarse mesh, simplices containing more points will be refined, and simplies containing fewer points will be retained. When we detect that the number of points (training input data) contained in all simplexes is not much different, the mesh will stop being refined.

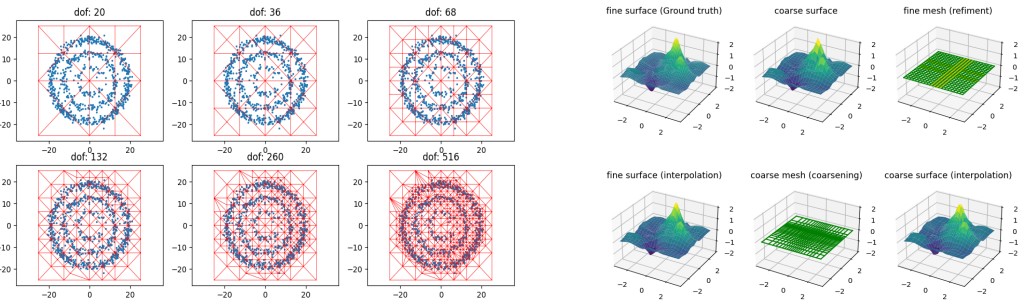

Figure 7: Left - Each triangle represents a 2-dimensional simplex, with increasing degrees of freedom indicating higher levels of refinement. Discrete points denote the raw data. Right - V-cycle transforms a uniform mesh into a multiscale mesh by inspecting residuals, redistributing nodes from flat regions to improve the mesh's representation capacity.

# E    ADDITIONAL EXPERIMENTS

We study the robustness of the LagEncoder on the ImageNet-1K validation set with one training epoch. The result is presented in Table 3. All methods follow the raw training recipes to ensure fair comparisons.

| Model | $d$ | $n$ | # Params | Acc@1 (%) | Acc@5 (%) |
|---|---|---|---|---|---|
| MobileNet-V2 | 32 | 8 | 0.298 M (+8.51%) | $71.920_{\pm.016}(+.045)$ | $90.302_{\pm.021}(+.016)$ |
| ResNet50 | 32 | 8 | 0.324 M (+1.27%) | $76.334_{\pm.054}(+.204)$ | $92.976_{\pm.024}(+.114)$ |
| ResNeXt50 | 32 | 8 | 0.324 M (+1.30%) | $77.796_{\pm.063}(+.178)$ | $93.653_{\pm.038}(+.178)$ |
| ViT-B-16 | 8 | 32 | 1.032 M (+1.19%) | $81.092_{\pm.012}(+.020)$ | $95.316_{\pm.004}(-.002)$ |

Table 3: Classification accuracy on the ImageNet-1K validation set and one epoch training, with all methods adhering to the raw training recipes. We report the changes in the number of model parameters and their ratio of changes and total. For each model, we conducted multiple experiments and show the accuracy as "$\text{mean}_{\pm\text{std}}(\text{mean} - \text{baseline})$".

| Metric $(n, d)$ | (4, 4) | (4, 8) | (4, 16) | (4, 32) | (8, 4) | (8, 8) | (8, 16) | (8, 32) |
|---|---|---|---|---|---|---|---|---|
| # Parameters | 36,868 | 69,636 | 135,172 | 266,244 | 73,736 | 139,272 | 270,344 | 532,488 |
| Acc@1 | 76.226 | 76.226 | **76.242** | 76.222 | 76.238 | 76.256 | 76.268 | **76.276** |
| Acc@5 | 92.952 | 92.960 | 92.948 | **92.972** | 92.964 | 92.954 | **92.966** | 92.952 |
| Speed (img/s) | 405.31 | 411.18 | 390.26 | **418.84** | 382.22 | 394.63 | **409.52** | 404.43 |
| Metric | (16, 4) | (16, 8) | (16, 16) | (16, 32) | (32, 4) | (32, 8) | (32, 16) | (32, 32) |
| # Parameters | 147,472 | 278,544 | 540,688 | 1,064,976 | 294,944 | 557,088 | 1,081,376 | 2,129,952 |
| Acc@1 | 76.272 | 76.244 | 76.248 | **76.284** | **76.298** | 76.272 | 76.248 | 76.294 |
| Acc@5 | 92.940 | 92.946 | **92.968** | 92.958 | 92.956 | 92.958 | **92.968** | 92.932 |
| Speed (img/s) | 393.19 | **394.86** | 379.72 | 380.30 | 393.34 | **414.39** | 379.72 | 368.94 |

Table 4: ResNet-50 Performance with different $n$ and $d$. The table explores the impact of varying $n$ (PCA output dimension) and $d$ (degrees of freedom) on the performance of the LagEncoder-based ResNet-50 model. Metrics include the number of additional parameters, top-1 (Acc@1) and top-5 (Acc@5) validation accuracy, and training speed (images per second). The baseline ResNet-50 has 25.6M parameters, Acc@1: 76.130%, and Acc@5: 92.862%.

Increasing $n$, $d$, and training epochs is an effective way to enhance model performance, as shown in Table 4. *Unlike the function fitting task, diminishing returns in performance improvements are observed in this case due to the finite cardinality of image and text datasets.*

LoRA and other PEFT methods perform exceptionally well in transfer learning when adapting to new datasets or domains with limited data or computational resources. However, they may not be ideal for tasks where fine-tuning models on the same dataset and have the following limitations.

1. When the dataset is fixed and does not differ significantly from the original dataset used for pre-training, PEFT methods may underperform compared to training from scratch.

2. When using LoRA or other PEFT methods, it's crucial to reset the training recipe (including batch size, optimizer, and use a small learning rate of $10^{-4} \sim 10^{-6}$).

3. Most PEFT methods are not general. Many Parameter-Efficient Fine-Tuning (PEFT) methods rely on model-specific and task-specific configurations. For example, IA3 and LoHA apply to Linear and Conv1D layers only.

We also compared LoRA with our method of transfer learning by learning the fusion of WikiText-2 and WikiText-103 datasets. As shown in Table 5, our method outperforms LoRA in this scenario.

| Model | Method | # Trainable Parameters | Acc | Loss | PPL | Seq/s |
|---|---|---|---|---|---|---|
| GPT2 | FT | 124,439,808 | 0.422 | 3.0756 | 21.6623 | 7.088 |
| | LoRA | 294,912 | 0.4221 | 3.0747 | 21.6431 | **13.048** |
| | LagEncoder | 202,566 | **0.4229** | **3.0702** | **21.5469** | 9.392 |
| GPT2-Medium | FT | 354,823,168 | 0.455 | 2.78 | 16.1183 | 3.376 |
| | LoRA | 393,216 | 0.4556 | 2.7783 | 16.0916 | 6.152 |
| | LagEncoder | 404,106 | **0.4558** | **2.7773** | **16.0751** | **6.592** |
| GPT2-Large | FT | 774,030,080 | 0.4721 | 2.6385 | 13.9927 | 1.712 |
| | LoRA | 737,280 | **0.4725** | **2.6347** | **13.9387** | 1.648 |
| | LagEncoder | 606,927 | 0.4724 | 2.6369 | 13.97 | **2.34** |

Table 5: Comparison of Fine-Tuning (FT), LoRA, and LagEncoder on Transfer Learning with Fused WikiText-2 and WikiText-103 Datasets. The table reports the number of trainable parameters, accuracy (Acc), loss, perplexity (PPL), and training throughput (sequences per second, Seq/s) for each method and model size. LagEncoder achieves competitive accuracy and perplexity while maintaining efficient parameter utilization and high training efficiency.

# F ADDITIONAL APPLICATIONS

## F.1 REGRESSION

As mentioned earlier, neural networks often struggle to fit multi-frequency datasets effectively. Therefore, our primary focus is comparing the LagEncoder-based model with traditional regressors. To evaluate the effectiveness and generalization of the LagEncoder-based model, we have devised four diverse datasets, each generated from distinct probability distributions:

1. $\mathbb{A}^1$: Generated from the distribution $\{(x,y) | X \sim U(-\pi, \pi), Y \sim \mathcal{N}(\sin x, \frac{1}{5} \cos^2 x)\}$, with 1000 training examples and 200 test examples. The LagEncoder-based model was trained with a learning rate of 0.1.

2. $\mathbb{B}^1$: Generated from the distribution $\{(x,y) | \frac{1}{X} \sim U(0.02, 1.0), Y \sim \mathcal{N}(\sin \frac{1}{x}, 0.01)\}$, with 1000 training examples and 200 test examples. We trained the LagEncoder-based model with a learning rate of 0.9.

3. $\mathbb{A}^2$: Generated from the distribution $\{(\boldsymbol{x},y) | \boldsymbol{X}_i \sim U(-\pi, \pi), Y_i \sim \mathcal{N}(\sin \boldsymbol{x}_i, \frac{1}{10} \cos^2 \boldsymbol{x}_i)$ , $Y = \frac{1}{2}(Y_1 + Y_2)\}$, with 7,500 training examples and 1,500 test examples. The LagEncoder-based model was trained with a learning rate of 0.1.

4. $\mathbb{B}^2$: Generated from the distribution $\{(\boldsymbol{x}, y)|\frac{1}{\boldsymbol{X}_i} \sim U(0.05, 0.5), Y_i \sim \mathcal{N}(\sin \frac{1}{\boldsymbol{x}_i}, 0.01), Y = \frac{1}{2}(Y_1 + Y_2)\}$, with 50,000 training examples and 10,000 test examples. The training utilized a learning rate of 0.9.

Table 6 displays the coefficient of determination ($R^2$) scores for the LagEncoder-based model and traditional regressors (Thiel, 1950; Cantzler, 1981; Zhang, 2004; Hilt & Seegrist, 1977; Stone, 1974; Jain et al., 2018; Murphy, 2012; Platt et al., 1999; Friedman, 2001; Breiman, 2001) across fitting the four datasets. The LagEncoder-based model consistently achieves high $R^2$ scores across all test sets, demonstrating the effectiveness of the InterpolationNet on both high-noise and multi-frequency datasets. Furthermore, the minimal gap between training and test set evaluations underscores the robustness of the LagEncoder-based model, indicating its capability of generalization.

| METHOD | $\mathbb{A}^1$ | | $\mathbb{B}^1$ | | $\mathbb{A}^2$ | | $\mathbb{B}^2$ | |
|---|---|---|---|---|---|---|---|---|
| OLS Linear | 0.037 | 0.042 | 0.963 | 0.951 | 0.085 | 0.092 | 0.984 | 0.984 |
| Theil-Sen | -44.7 | -54.4 | 0.958 | 0.946 | -0.41 | -3.81 | 0.982 | 0.982 |
| RANSAC | -1.21 | -1.43 | 0.963 | 0.951 | -27.0 | -27.1 | 0.983 | 0.983 |
| Huber | 0.036 | 0.041 | 0.962 | 0.949 | 0.085 | 0.092 | 0.984 | 0.984 |
| Ridge | 0.031 | 0.038 | 0.963 | 0.951 | 0.055 | 0.061 | 0.984 | 0.984 |
| RidgeCV | 0.037 | 0.042 | 0.963 | 0.951 | 0.085 | 0.092 | 0.984 | 0.984 |
| SGD | 0.009 | 0.01 | 0.962 | 0.95 | 0.005 | 0.004 | 0.983 | 0.983 |
| KRR | 0.0036 | 0.04 | 0.97 | 0.962 | 0.056 | 0.051 | 0.993 | 0.992 |
| SVR | 0.11 | 0.101 | 0.97 | 0.962 | 0.29 | 0.308 | 0.992 | 0.992 |
| Gradient Boosting | 0.964 | 0.962 | 0.98 | 0.96 | 0.989 | 0.988 | 0.992 | 0.99 |
| Random Forests | 1.0 | 0.999 | 0.995 | 0.945 | 1.0 | 0.999 | 0.999 | 0.99 |
| Voting | 0.852 | 0.852 | 0.942 | 0.917 | 0.869 | 0.868 | 0.951 | 0.946 |
| Net | 1.0 | 1.0 | 0.971 | 0.963 | 0.999 | 0.999 | 0.992 | 0.992 |

Table 6: A comprehensive comparison between the LagEncoder-based model and traditional regressors. The left half of each paired column displays the training $R^2$ score, while the right half showcases the corresponding test $R^2$ score.

### F.2 FITTING HIGH-NOISE DATASET

In this section, we conduct the LagEncoder-based model on fitting the dataset $\mathbb{A} = \{(x, y)|X \sim U(-4, 4), Y \sim \mathcal{N}(\sin x, 0.2 \cos^2 x)\}$. A comprises 6000 examples, with 5000 for training and 1000 for testing. Fig. 8 shows the training progress.

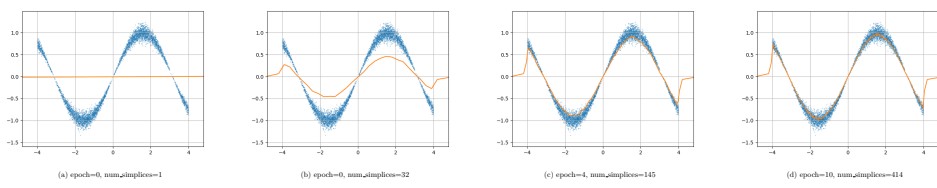

(a) epoch=0, num_simplices=1  (b) epoch=0, num_simplices=32  (c) epoch=4, num_simplices=145  (d) epoch=10, num_simplices=414

Figure 8: Blue dots represent the training set, while the orange curve represents the network.

### F.3 FITTING MULTI-FREQUENCY DATASET

In this section, we conduct the LagEncoder-based model on fitting the dataset $\mathbb{A} = \{(x, y)|\frac{1}{x} \sim U(0.02, 0.5), y = \sin \frac{1}{x}\}$. A comprises 6000 examples, with 5000 for training and 1000 for testing. Fig. 9 illustrates the training progress. Remarkably, after just 4 epochs of training, the neural network outputs closely approximate the target values. By the 32nd epoch's conclusion, the neural network outputs and target values are nearly indistinguishable.

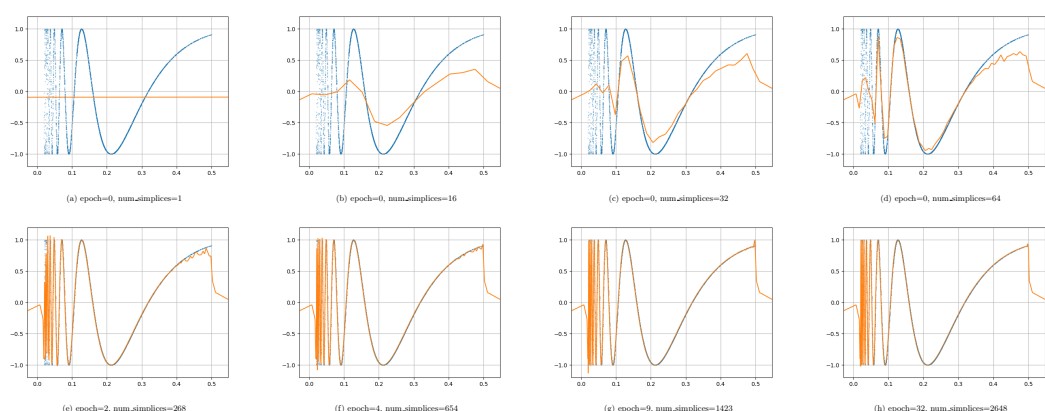

(a) epoch=0, num_simplices=1    (b) epoch=0, num_simplices=16    (c) epoch=0, num_simplices=32    (d) epoch=0, num_simplices=64

(e) epoch=2, num_simplices=268    (f) epoch=4, num_simplices=654    (g) epoch=9, num_simplices=1423    (h) epoch=32, num_simplices=2648

Figure 9: Blue dots represent the training set, while the orange curve represents the network.

### F.4 FIT A VECTOR-VALUED FUNCTION

In this instance, we utilize the LagEncoder-based model to fit spherical harmonics. Our dataset denoted as $\mathbb{A} = \{(\boldsymbol{x}, \boldsymbol{y}) | \boldsymbol{x} = (\theta, \phi), \boldsymbol{y} = (\mathrm{Real}(Y_4^2(\theta, \phi)), \mathrm{Imag}(Y_4^2(\theta, \phi))), \Theta \sim U(0, 2\pi), \Phi \sim U(0, \pi)\}$, comprises 48,000 examples, with 40,000 allocated for training and an additional 8,000 for testing. Fig. 10 shows the training progress.

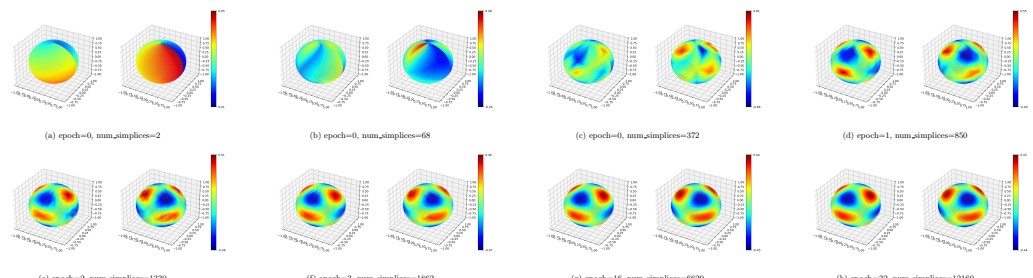

(a) epoch=0, num_simplices=2    (b) epoch=0, num_simplices=68    (c) epoch=0, num_simplices=372    (d) epoch=1, num_simplices=850

(e) epoch=2, num_simplices=1330    (f) epoch=3, num_simplices=1862    (g) epoch=16, num_simplices=6620    (h) epoch=32, num_simplices=12160

Figure 10: In each block, the left panel represents the real part of our model output, while the right panel represents the imaginary part of the model output.

### F.5 SOLVE PDES

In this section, we utilize the LagEncoder-based model to address the following partial differential equations (PDEs):

$$\begin{cases} \Delta u + (u - \beta)^2 = (\alpha \cos x \sin y - 1)^2 + 1, & (x, y) \in \Omega; \\ u = \beta, & (x, y) \in \partial\Omega. \end{cases}$$

Here, $\Omega = [0, 1] \times [0, 1]$. We construct a dataset that takes $(\alpha, \beta)$ as input data and assigns the

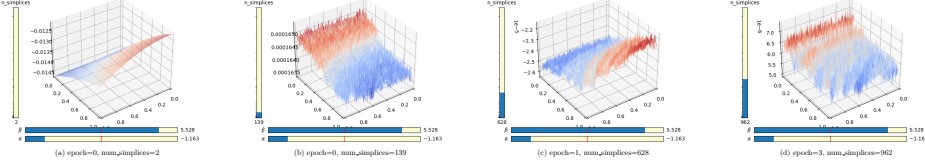

(a) epoch=0, num_simplices=2    (b) epoch=0, num_simplices=139    (c) epoch=1, num_simplices=628    (d) epoch=3, num_simplices=962

Figure 11: Residual - The gap between the exact solution and the model output.

corresponding numerical solution of the PDEs as the target output. This dataset comprises 12,000 examples, with $\alpha$ randomly selected from the distribution $U(-\pi/2, \pi/2)$ and $\beta$ randomly chosen from the distribution $U(0, 2\pi)$. We then split the dataset into two parts: 10,000 for training and 2,000 for testing. Fig. 11 illustrates how well the network predicts the exact solution.

