# OpenReview forum: "LagEncoder: A Non-Parametric Method for Representation Learning"
_ICLR.cc/2025/Conference — Submitted to ICLR 2025_

### Official Review · Reviewer_6NPx · 2024-11-02

**Soundness:** 3
**Presentation:** 3
**Contribution:** 2
**Rating:** 6
**Confidence:** 2

**Summary:**

This paper proposes using the Finite Element Method (FEM) for training-free feature extraction, particularly using the Lagrange basis function. Furthermore, the paper re-derives the Lagrange basis to exploit parallelism in a deep learning setting. The proposed LagEncoder is universal and is demonstrated on regression, image classification, text classification. Since LagEncoder can have high computational demands when dealing with high dimensional data, the paper demonstrates how it can be incorporated as a parameter efficient fine-tuning method. Scaling laws show that LagEncoder can outperform purely scaling model size.

**Strengths:**

* To the best of my knowledge, application of the Finite Element Method for representation learning is a novel contribution.
* Other contributions of the paper include tricks to adapt the Lagrange basis to a deep learning setting, e.g., a re-derivation that allows parallelism, incorporating it into PEFT modules.
* Scaling laws show that incorporating LagEncoder with a negligible amount of parameters can reach the same performance as a scaled-up version of the base model.
* Experiments are conducted for multiple domains: regression, vision, text classification. One particularly compelling result is matching a 6.13 million parameter word2vec model using only 256 parameters.

**Weaknesses:**

* The biggest limitation is addressed by the paper itself. It is extremely expensive to use LagEncoder directly on high dimensional data, which "restricts its direct application to large-scale datasets."
* For the NLP task, LagEncoder is compared against word2vec which is more than ten years old, limiting the relevancy of this evaluation. Can LagEncoder be incorporated in modern language models?
* For the vision tasks, the improvements from LagEncoder seem negligible with a fraction of a percent improvement.
* The paper lacks comparison against other PEFT methods such as LoRA.

**Questions:**

* Can LagEncoder be incorporated in modern language models?
* How does LagEncoder compare against the widely used PEFT method LoRA? The PCA and residual mentioned in Section 2.3 are highly reminiscent of LoRA.

---

> ### Author Response · Authors · 2024-11-24
> **Response to reviewer 6NPx (1/3)**
>
> We sincerely appreciate your thorough review of our paper and the concerns you raised regarding our experiments. Here is our response to your questions:
>
> # Weakness
>
> >The biggest limitation is addressed by the paper itself. It is extremely expensive to use LagEncoder directly on high dimensional data, which "restricts its direct application to large-scale datasets."
>
> Yes, we have already addressed this limitation using Algorithm 2. In our experiments on the ImageNet-1K and WikiText datasets, the input dimension of LagEncoder increased to 2,048, and the output dimension increased to 50,257.
>
> >For the NLP task, LagEncoder is compared against word2vec which is more than ten years old, limiting the relevancy of this evaluation. Can LagEncoder be incorporated in modern language models?
>
> Thank you for your feedback. Before introducing the application of LagEncoder on LLMs, we needed to introduce three key limitations of other existing PEFT methods. This is why such experiments were not included in the earlier version of our paper.
>
> - Dependence on Transfer Learning: Methods like LoRA struggle to outperform pre-trained models on the same or similar datasets, as no domain adaptation is needed. Additional training data is often required for these methods to be effective, as shown in the following Table. Their strength lies mainly in transfer learning scenarios. However, when trained from scratch on the raw dataset, they fail to outperform pre-trained models (see Section 3.3, Table 2).
> - Sensitive Training Requirements: PEFT methods require specific training recipes, such as small learning rates. Without these, performance often deteriorates from the first epoch.
> - Task and Architecture Limits: Many PEFT methods, like LoHA [3] and IA3 [4], are restricted to specific tasks or architectures, such as Conv1D or linear layers.
>
> We compare fine-tuning (FT), LoRA, and LagEncoder on WikiText benchmarks [5] using GPT2 [6] for causal language modeling and RoBERTa [7] for masked language modeling. Pre-trained models and the default random seed (42) of the HuggingFace Transformers library [8] are used. A reduced learning rate ($\le 10^{-5}$) is applied for LoRA to prevent performance degradation in these experiments. As shown in the following table, our method overcomes the mentioned challenges, performing well even without extra training data.
>
> In these table. We report the number of trainable parameters, perplexity (PPL, lower is better), and training throughput (sequences per second) for language modeling tasks. For a fair comparison, we adjusted the number of parameters of our method to be similar to those of LoRA.
>
> **Wikitext-2-raw-v1 dataset**
>
> | Model        | Method     | # Params    | Acc    | Loss   | Perplexity | Seq/s      |
> |--------------|------------|-------------|--------|--------|------------|------------|
> | GPT2         |            | 124,439,808 | 0.422  | 3.0756 | 21.6623    | 7.36       |
> | GPT2         | LoRA       | 294,912     | 0.422  | 3.0754 | 21.6582    | 13.048     |
> | GPT2         | LagEncoder | 202,566     | 0.4227 | 3.0726 | 21.597     | 9.76       |
> | GPT2-Medium  |            | 354,823,168 | 0.455  | 2.78   | 16.1183    | 3.376      |
> | GPT2-Medium  | LoRA       | 393,216     | 0.455  | 2.78   | 16.1183    | 6.00       |
> | GPT2-Medium  | LagEncoder | 404,106     | 0.4553 | 2.7794 | 16.1086    | 5.832      |
> | GPT2-Large   |            | 774,030,080 | 0.4721 | 2.6385 | 13.9927    | 1.712      |
> | GPT2-Large   | LoRA       | 737,280     | 0.472  | 2.6389 | 13.9973    | 1.688      |
> | GPT2-Large   | LagEncoder | 606,927     | 0.4723 | 2.6388 | 13.9964    | 3.016      |
> | Roberta-base |            | 124,697,433 | 0.7258 | 1.2903 | 3.6337     | 9.820      |
> | Roberta-base | LoRA       | 294,912     | 0.7257 | 1.2905 | 3.6347     | 9.676      |
> | Roberta-base | LagEncoder | 303,128     | 0.7257 | 1.2905 | 3.6346     | 15.392     |

---

> ### Author Response · Authors · 2024-11-24
> **Response to reviewer 6NPx (2/3)**
>
> **Wikitext-103-raw-v1 dataset**
>
> | Model        | Method     | # Params    | Acc    | Loss   | Perplexity | Seq/s      |
> |--------------|------------|-------------|--------|--------|------------|------------|
> | GPT2         |            | 124,439,808 | 0.4231 | 3.0645 | 21.4234    | 6.984      |
> | GPT2         | LoRA       | 294,912     | 0.4232 | 3.0643 | 21.4188    | 12.896     |
> | GPT2         | LagEncoder | 202,566     | 0.4237 | 3.2036 | 21.3401    | 9.584      |
> | GPT2-Medium  |            | 354,823,168 | 0.4572 | 2.7657 | 15.89      | 3.496      |
> | GPT2-Medium  | LoRA       | 393,216     | 0.4571 | 2.7655 | 15.8873    | 6.288      |
> | GPT2-Medium  | LagEncoder | 404,106     | 0.4573 | 2.7643 | 15.8674    | 6.656      |
> | GPT2-Large   |            | 774,030,080 | 0.4734 | 2.6281 | 13.8468    | 1.584      |
> | GPT2-Large   | LoRA       | 737,280     | 0.4734 | 2.6282 | 13.8486    | 1.668      |
> | GPT2-Large   | LagEncoder | 606,927     | 0.4733 | 2.6279 | 13.8443    | 2.516      |
> | Roberta-base |            | 124,697,433 | 0.7252 | 1.2924 | 3.6415     | 10.052     |
> | Roberta-base | LoRA       | 294,912     | 0.725  | 1.2927 | 3.6425     | 9.852      |
> | Roberta-base | LagEncoder | 303,128     | 0.7251 | 1.2924 | 3.6414     | 12.368     |
>
> We also compared LoRA with our method of transfer learning by learning the fusion of WikiText-2 and WikiText-103 datasets. As shown in Table 6, our method outperforms LoRA in this scenario.
>
>
> | Model         | Method     | # Trainable Parameters | Acc    | Loss   | PPL        | Seq/s    |
> |---------------|------------|------------------------|--------|--------|------------|----------|
> | GPT2          | FT         | 124,439,808            | 0.422  | 3.0756 | 21.6623    | 7.088    |
> | GPT2              | LoRA       | 294,912                | 0.4221 | 3.0747 | 21.6431    | 13.048   |
> | GPT2              | LagEncoder | 202,566                | 0.4229 | 3.0702 | 21.5469    | 9.392    |
> | GPT2-Medium   | FT         | 354,823,168            | 0.455  | 2.78   | 16.1183    | 3.376    |
> | GPT2-Medium              | LoRA       | 393,216                | 0.4556 | 2.7783 | 16.0916    | 6.152    |
> |  GPT2-Medium             | LagEncoder | 404,106                | 0.4558 | 2.7773 | 16.0751    | 6.592    |
> | GPT2-Large    | FT         | 774,030,080            | 0.4721 | 2.6385 | 13.9927    | 1.712    |
> | GPT2-Large              | LoRA       | 737,280                | 0.4725 | 2.6347 | 13.9387    | 1.648    |
> |  GPT2-Large             | LagEncoder | 606,927                | 0.4724 | 2.6369 | 13.97      | 2.34     |
>
> >For the vision tasks, the improvements from LagEncoder seem negligible with a fraction of a percent improvement.
>
> Yes, improving the performance of a well-trained model on the raw dataset is challenging. Here are some comparisons:
>
> - **Data Augmentation:** Cutout, one of the most popular data augmentation method, increases the accuracy of ResNet-50 on ImageNet-1K by 0.8%. However, it requires a significant redesign of the training recipe and an increase in training epochs from 90 to 200 [9]. In comparison, our method provides similar performance improvements to those achieved by the vertical image translation augmentation method.
>
> - **Transfer/Continual Learning:** This method requires continuously adding new data to improve the model’s performance.
>
> - **Knowledge Distillation:** This approach requires a larger teacher model and still necessitates a redesign of the training recipe, along with training the model from scratch.
>
> In conclusion, our method does not have these constraints but still delivers quick improvements to the model. Additionally, it can be easily integrated with the techniques mentioned above.
>
> >The paper lacks comparison against other PEFT methods such as LoRA.
>
> Thank you for your comment. We did not include a comparison with other PEFT methods because our approach focuses on improving model performance on the raw dataset, whereas other PEFT methods rely on pre-trained models and training on new datasets. As shown in the tables in this response and our reply to reviewer **6J2S**, our method outperforms LoRA. Additionally, our approach does not require a very small learning rate and is neither task- nor model-specific.
>
>
> # Questions
>
> >Can LagEncoder be incorporated in modern language models?
>
> Yes, we present our experimental results in response to Weakness 2.
>
> >How does LagEncoder compare against the widely used PEFT method LoRA? The PCA and residual mentioned in Section 2.3 are highly reminiscent of LoRA.
>
> In our response to Weakness 2, we demonstrate the advantages of our model, showing that it outperforms LoRA in both tuning on the raw dataset and transfer learning tasks.

---

> ### Author Response · Authors · 2024-11-24
> **Response to reviewer 6NPx (3/3)**
>
> # Reference
>
> [3] Nam Hyeon-Woo, Moon Ye-Bin, and Tae-Hyun Oh. Fedpara: Low-rank hadamard product for communication-efficient federated learning. arXiv preprint arXiv:2108.06098, 2021.
>
> [4] Haokun Liu, Derek Tam, Mohammed Muqeeth, Jay Mohta, Tenghao Huang, Mohit Bansal, and Colin A Raffel. Few-shot parameter-efficient fine-tuning is better and cheaper than in-context learning. Advances in Neural Information Processing Systems, 35:1950–1965, 2022.
>
> [5] Stephen Merity, Caiming Xiong, James Bradbury, and Richard Socher. Pointer sentinel mixture models. arXiv preprint arXiv:1609.07843, 2016.
>
> [6] Alec Radford, Jeffrey Wu, Rewon Child, David Luan, Dario Amodei, Ilya Sutskever, et al. Language models are unsupervised multitask learners. OpenAI blog, 1(8):9, 2019.
>
> [7] Yinhan Liu. Roberta: A robustly optimized bert pretraining approach. arXiv preprint arXiv:1907.11692, 364, 2019.
>
> [8] Thomas Wolf, Lysandre Debut, Victor Sanh, Julien Chaumond, Clement Delangue, Anthony Moi, Pierric Cistac, Tim Rault, Remi Louf, Morgan Funtowicz, et al. Transformers: State-of-the-art natural language processing. In Proceedings of the 2020 conference on empirical methods in natural language processing: system demonstrations, pp. 38–45, 2020.
>
> [9] Terrance DeVries and Graham W Taylor. Improved regularization of convolutional neural networks with cutout. arXiv preprint arXiv:1708.04552, 2017.
>
> # Updates
>
> We have also included updated code in the supplemental materials.
>
> - To reproduce our results, please execute the following command:
> ```bash
> cd language-modeling
> bash run_clm.sh
> bash run_clm_transfer.sh
> bash run_mlm.sh
> bash run_mlm_transfer.sh
> ```
>
> - To quickly review the training log, refer to `language-modeling/clm_output.log`, `language-modeling/clm_transfer_output.log`, `language-modeling/mlm_output.log`, and `language-modeling/mlm_transfer_output.log`.

---

> > ### Comment · Reviewer_6NPx · 2024-11-24
> >
> > Thank you for your extensive rebuttal. It has answered my questions, thus I will raise my score. I am fairly uncertain about this domain, so I am willing to defer to the AC and reviewers who may have more expertise here.

---

> ### Author Response · Authors · 2024-11-24
> **Response to reviewer 6NPx**
>
> Thank you so much for your recognition. We will try our best to provide more detailed experimental results within the remaining limited time.

---

### Official Review · Reviewer_6J2S · 2024-11-02

**Soundness:** 3
**Presentation:** 1
**Contribution:** 2
**Rating:** 5
**Confidence:** 4

**Summary:**

The paper presents LagEncoder, a nonparametric, training-free feature extraction method based on finite element basis
functions.
The encoder can be combined with various model architecture withr reasonable performances.
The experiments on the ImageNet dataset demonstrate that pre-trained models using
LagEncoder achieve performance improvements within just one training epoch.

**Strengths:**

The paper idea is novel and the results are encouraging.

**Weaknesses:**

The writing of the paper is super bad. It is very hard to track different symbols and the symbols sometimes are wrong.
1. In Eq.(4), the paper introduces p but never defined before or at the equation.
2. For matrix T, the meaning of the values in the matrix are never defined. In my understanding, each column of the matrix should describe the seven simplices relationship to the corresponding nodes.
3. The relationship of i,j,k in Eq.(5) is not clearly defined. It takes me time to figure out the meaning of n, n_t and d.
4. In Algorithm 1, you introduced completely new symbol definitions compared to previous versions, which further decreases the readability of the paper.
5. In algorithm 2, what is v(i) in compute loss step? I would guess it is x(i).

**Questions:**

1. For the benchmark in NLP, what is the performance if the freedom n of LagEncoder increased?
2. The paper claims interpretability, which I completely did not see from experiments. I think some attributions to the input is the interpretability.  The experiments in 3.1.2 is unconvincing.
3. The few epochs extra training is very impressive. However, can this method be combined with original architecture and directly train from scratch? Will that also improve the performance?
4. For the experiments in table 1, what is the performance change if we increase d and n?

---

> ### Author Response · Authors · 2024-11-23
> **Response to reviewer 6J2S (1/3)**
>
> We sincerely appreciate your thorough review of our paper and the concerns you raised regarding our experiments. Here is our response to your questions:
>
> # Weakness
>
> >In Eq.(4), the paper introduces $p$ but never defined before or at the equation.
>
> This is a standard concept in the **Domain Decomposition Method/Delaunay Triangulation**. In this paper, the definition of $p$ first appears in line 139, where it represents the grid node. In Eq. (4), it refers to the one-dimensional case. For visualization, please see Fig. 1 (left and middle).
>
> >For matrix $T$, the meaning of the values in the matrix are never defined. In my understanding, each column of the matrix should describe the seven simplices relationship to the corresponding nodes.
>
> This is a standard concept in the **Domain Decomposition Method/Delaunay Triangulation**. In this paper, the definition of matrix $T$ first appears in line 142, where it represents the indices of nodes constituting the triangles/simplices within the triangulation.
>
> >The relationship of $i,j,k$ in Eq.(5) is not clearly defined. It takes me time to figure out the meaning of $n$, $n_t$ and $d$.
>
> These are standard concepts in the **Domain Decomposition Method/Delaunay Triangulation**.
> - $i,j,k$ are indices of a 3-dimensional array and do not require explicit definitions.
> - The definitions of $n$ and $d$ first appear in line 139, where $n$ represents the number of grid nodes and $d$ denotes the spatial dimension.
> - The definition of $n_t$ first appears in line 165, where it represents the number of simplices in the multiscale mesh.
>
> >In Algorithm 1, you introduced completely new symbol definitions compared to previous versions, which further decreases the readability of the paper.
>
> `In this paper, we do not introduce any new geometrical concepts.` All statements in Algorithm 1 are constructed using standard academic terms from the **Domain Decomposition Method** / **Delaunay Triangulation**.
>
> >In algorithm 2, what is $v(i)$ in compute loss step? I would guess it is $x(i)$.
>
> In this paper, the definition of $v(i)$ first appears in line 238,  it represents the reduced-dimensionality feature and it is the input of SigmoidNorm layer. `We also provide animations to better illustrate Algorithms 1 and 2 (see Supplementary Material: LagEncoder/mesh_refinement.gif and LagEncoder/to_equal_freq_binning.mp4).`

---

> ### Author Response · Authors · 2024-11-23
> **Response to reviewer 6J2S (2/3)**
>
> # Questions
> >For the benchmark in NLP, what is the performance if the freedom n of LagEncoder increased?
>
> Increasing $n$ and $d$ improves model performance (see Appendix E, Table 4). Additionally, we conducted experiments on GPT2 and RoBERTa models using the WikiText datasets for causal and masked language modeling tasks. For further details, please refer to our response to reviewer **6NPx**.
>
> | Model    | n  | d  | Params   | Acc@1  | Acc@5  | img/s  |
> |----------|----|----|----------|--------|--------|--------|
> | resnet50 |    |    | 25.6M    | 76.130 | 92.862 |        |
> | resnet50 | 4  | 4  | 36,868   | 76.226 | 92.952 | 405.31 |
> | resnet50 | 4  | 8  | 69,636   | 76.226 | 92.960 | 411.18 |
> | resnet50 | 4  | 16 | 135,172  | 76.242 | 92.948 | 390.26 |
> | resnet50 | 4  | 32 | 266,244  | 76.222 | 92.972 | 418.84 |
> | resnet50 | 8  | 4  | 73,736   | 76.238 | 92.964 | 382.22 |
> | resnet50 | 8  | 8  | 139,272  | 76.256 | 92.954 | 394.63 |
> | resnet50 | 8  | 16 | 270,344  | 76.268 | 92.966 | 409.52 |
> | resnet50 | 8  | 32 | 532,488  | 76.276 | 92.952 | 404.43 |
> | resnet50 | 16 | 4  | 147,472  | 76.272 | 92.940 | 393.19 |
> | resnet50 | 16 | 8  | 278,544  | 76.244 | 92.946 | 394.86 |
> | resnet50 | 16 | 16 | 540,688  | 76.248 | 92.968 | 379.72 |
> | resnet50 | 16 | 32 | 1,064,976| 76.284 | 92.958 | 380.30 |
> | resnet50 | 32 | 4  | 294,944  | 76.298 | 92.956 | 393.34 |
> | resnet50 | 32 | 8  | 557,088  | 76.272 | 92.958 | 414.39 |
> | resnet50 | 32 | 16 | 1,081,376| 76.248 | 92.968 | 379.72 |
> | resnet50 | 32 | 32 | 2,129,952| 76.294 | 92.932 | 368.94 |
>
> This table explores the impact of varying $n$ (PCA output dimension) and $d$ (degrees of freedom) on the performance of the LagEncoder-based ResNet-50 model. Metrics include the number of additional parameters, top-1 (Acc@1) and top-5 (Acc@5) validation accuracy, and training speed (images per second). The baseline ResNet-50 has 25.6M parameters, Acc@1: 76.130\%, and Acc@5: 92.862\% on ImageNet-1K.
>
> >The paper claims interpretability, which I completely did not see from experiments. I think some attributions to the input is the interpretability. The experiments in 3.1.2 is unconvincing.
>
> Let us refer to prior works [1, 2] to examine how they introduce interpretability. Models often adhere to an empirical scaling law. When this formula can be derived from a theoretical foundation, the model becomes interpretable. In Section 3.1.2, we present quantitative experimental results supporting the deduced error-bound formula. **While the error-bound formula originates from the universal approximation theorem, it is now commonly referred to as the scaling law in the field of NLP.**
>
> > The few epochs extra training is very impressive. However, can this method be combined with original architecture and directly train from scratch? Will that also improve the performance?
>
> Yes, increasing the training epochs (1–4 epochs) further improves model performance. However, training from scratch is unnecessary. The following table provides a time efficiency comparison.
>
> | Model           | Method       | Params   | Acc@1  | Acc@5  | img/s   | Wall-Clock Time |
> |------------------|--------------|----------|--------|--------|---------|-----------------|
> | mobilenet_v2*   |              | 3.5M     | 71.878 | 90.286 |         | 16.5h           |
> | mobilenet_v2     | LagEncoder   | 328,720  | 71.934 | 90.268 | 685.17  | 32.2m           |
> | resnet50*        |              | 25.6M    | 76.130 | 92.862 |         | 2d 1h 15m       |
> | resnet50         | LagEncoder   | 540,688  | 76.274 | 92.932 | 411.50  | 40.7m           |
> | resnext50_32x4d* |              | 25.0M    | 77.618 | 93.698 |         | 3d 1h 30m       |
> | resnext50_32x4d  | LagEncoder   | 135,172  | 77.650 | 93.672 | 315.33  | 51.15m          |
> | vit_b_16*        |              | 86.6M    | 81.072 | 95.318 |         | 3d 3h 20m       |
> | vit_b_16         | LagEncoder   | 67,076   | 81.082 | 95.316 | 316.31  | 56.9m           |
>
> PEFT-LagEncoder demonstrates superior performance compared to train model from scratch, offering competitive trainable parameters, validation accuracy, training throughput (images per second), and total
> training time (five epochs).
>
> \* indicates numbers published in TorchVision: `MobileNet_V2_Weights.IMAGENET1K_V1`, `ResNet50_Weights.IMAGENET1K_V1`, `ResNeXt50_32X4D_Weights.IMAGENET1K_V1`, and `ViT_B_16_Weights.IMAGENET1K_V1`.
>
> > For the experiments in table 1, what is the performance change if we increase $d$ and $n$?
>
> Increasing $n$ and $d$ improves model performance (see Appendix E, Table 4).
>
>
> # Reference
>
> [1] Jared Kaplan, Sam McCandlish, Tom Henighan, Tom B Brown, Benjamin Chess, Rewon Child, Scott Gray, Alec Radford, Jeffrey Wu, and Dario Amodei. Scaling laws for neural language models. arXiv preprint arXiv:2001.08361, 2020.
>
> [2] Ziming Liu, Yixuan Wang, Sachin Vaidya, Fabian Ruehle, James Halverson, Marin Soljaci c, Thomas Y Hou, and Max Tegmark. Kan: Kolmogorov-arnold networks. arXiv preprint arXiv:2404.19756, 2024.

---

> ### Author Response · Authors · 2024-11-24
> **Response to reviewer 6J2S (3/3)**
>
> # Updates
>
> We have also included updated code in the supplemental materials.
>
> - To reproduce our results, please execute the following command:
> ```bash
> cd vision/PEFT-LagEncoder-full-trained
> bash run.sh
> ```
>
> - To quickly review the training log, refer to vision/PEFT-LagEncoder-full-trained/output.log.

---

### Official Review · Reviewer_eSdV · 2024-11-03

**Soundness:** 2
**Presentation:** 2
**Contribution:** 2
**Rating:** 5
**Confidence:** 4

**Summary:**

This paper introduces LagEncoder, a non-parametric, training-free feature extraction method based on Lagrange basis function.

**Strengths:**

The method have some empirical success in terms of regression and NLP and CV tasks.

**Weaknesses:**

1. The figure 4 is really hard to see.
2. The Computer vision results imporvement is really minor. Considering the extra computation it needed, it doesn't supervise me there is some improvement
3. The paper doesn't provide a good reason why I want to use this methods.

**Questions:**

1. If you still need a trainable backbone, how can you have strong mathematical explainability? Or do you believe your method have better explainability than a trainable linear layer?

---

> ### Author Response · Authors · 2024-11-24
> **Response to reviewer eSdV**
>
> We sincerely appreciate your thorough review of our paper and the concerns you raised regarding our experiments. Here is our response to your questions:
>
>
> # Weakness
>
> >The figure 4 is really hard to see.
>
> We modified the colors of the scatter plot on this figure.
>
> >The Computer vision results imporvement is really minor. Considering the extra computation it needed, it doesn't supervise me there is some improvement
>
> In the PEFT mode, our experiments show that ResNet-50 achieves a 0.2% improvement by adding 0.3M-1M parameters, and the entire training phase can be completed within 40 minutes (five training epochs) on four RTX A6000 GPUs (see our response to reviewer **6J2S**). In terms of rewarding accuracy improvements, it performs as well as the vertical image translation data augmentation method. However, our method is not constrained by redesigning the training recipe. Additionally, our method can improve model performance on the raw dataset without requiring extra training data, distinguishing it from other transfer learning methods. Finally, there is no conflict in integrating our method with the other approaches.
>
> # Questions
>
> >If you still need a trainable backbone, how can you have strong mathematical explainability? Or do you believe your method have better explainability than a trainable linear layer?
>
> Let us refer to prior works [1, 2] to examine how they introduce interpretability. Models often adhere to an empirical scaling law. When this formula can be derived from a theoretical foundation, the model becomes interpretable. In Section 3.1.2, we present quantitative experimental results supporting the deduced error-bound formula. **While the error-bound formula originates from the universal approximation theorem, it is now commonly referred to as the scaling law in the field of NLP.** Additionally, the backbone and heads of the pre-trained model are frozen in our PEFT method (see Fig. 3 (left)).
>
>
> # Reference
>
> [1] Jared Kaplan, Sam McCandlish, Tom Henighan, Tom B Brown, Benjamin Chess, Rewon Child, Scott Gray, Alec Radford, Jeffrey Wu, and Dario Amodei. Scaling laws for neural language models. arXiv preprint arXiv:2001.08361, 2020.
>
> [2] Ziming Liu, Yixuan Wang, Sachin Vaidya, Fabian Ruehle, James Halverson, Marin Soljaci c, Thomas Y Hou, and Max Tegmark. Kan: Kolmogorov-arnold networks. arXiv preprint arXiv:2404.19756, 2024.

---

> ### Comment · Reviewer_eSdV · 2024-11-27
>
> Thanks for the author's response. I adjusted my rating to 5 for the new manuscript.

---

### Meta-Review · Area_Chair_URtJ · 2024-12-25

**Metareview:**

The authors introduce LagEncoder, a non-parametric training-free feature extraction method based on finite element basis functions. One of the key motivations is to show that replacing NNs with standard interpolation methods isn't appropriate for fitting high-frequency regions. For example, KAN, which approximates the latent function using a linear combination of spline basis functions distributed on a uniform grid, is likely to underperform on high-frequency regions.

The presented method of using finite element basis indeed seems novel and promising. However, it was not easy to read the paper (writing requires a bit of work), the improvements shown are marginal (PEFT version), and there weren't enough arguments to convince why this work is useful in general.

Vanilla LagEncoder is not practically useful in high-dimensional data as the output dimension of LagEncoder grows factorially as the data dimension increases. The authors did suggest a fix to this problem which involved using it in combination as a parameter-efficient fine-tuning module (PEFT). This part of the paper (section 2.3) is crucial as it allows using high-dimensional real-world data, however, the rationale and validity behind PEFT, the arguments and assumptions used for the same are not clear and convincing enough. For example, lines 211:215 (`... if the pretrained model already achives high test accuracy ... pre-trained model's feaatures is likely reversible ...`). A mathematical view to this would be helpful or perhaps a few relevant references here. Additionally, as mentioned, PEFT-LagEncoder provided marginal improvements (Table 2, Table 5, and results during rebuttal).

Therefore, though the presented method and the direction is promising, I think this work requires a major rewrite for better readability and more convincing experiments (better performance gain in the PEFT setting, interpretability experiments, etc.).

**Additional Comments On Reviewer Discussion:**

- Most questions during the rebuttal revolved around (1) the practical utility of the proposed method (e.g, in high-dimensional data); (2) marginal improvement (PEFT version); (3) lack of comparisons w.r.t. other PEFT methods; and (4) relevance behind some assumptions.
- Authors did provide several replies to such comments for example (1) improved Fig 4 (2) new results using GPT2 and RoBERTA models on WikiText datasets etc., and we appreciate their effort. However, the paper would benefit greatly from better experimental results (as suggested during the rebuttal), careful rewriting, and solid thoughts on its practical utility including interpretability.

---

### Decision · Program_Chairs · 2025-01-22

Reject